# Gut microbiota aggravates neutrophil extracellular traps-induced pancreatic injury in hypertriglyceridemic pancreatitis

Guanqun Li[1,2,6], Liwei Liu[1,2,6], Tianqi Lu[1,2], Yuhang Sui[1,2], Can Zhang[1,2], Yongwei Wang[1], Tao Zhang[1], Yu Xie[1], Peng Xiao[1], Zhongjie Zhao[1], Chundong Cheng[1], Jisheng Hu[1,2], Hongze Chen[1,2], Dongbo Xue[1], Hua Chen[1], Gang Wang[1], Rui Kong[1,2], Hongtao Tan[1], Xuewei Bai[1], Zhibo Li[1], Florencia McAllister [3,4,5], Le Li [1,2] ✉ & Bei Sun [1,2] ✉

Hypertriglyceridemic pancreatitis (HTGP) is featured by higher incidence of complications and poor clinical outcomes. Gut microbiota dysbiosis is associated with pancreatic injury in HTGP and the mechanism remains unclear. Here, we observe lower diversity of gut microbiota and absence of beneficial bacteria in HTGP patients. In a fecal microbiota transplantation mouse model, the colonization of gut microbiota from HTGP patients recruits neutrophils and increases neutrophil extracellular traps (NETs) formation that exacerbates pancreatic injury and systemic inflammation. We find that decreased abundance of *Bacteroides uniformis* in gut microbiota impairs taurine production and increases IL-17 release in colon that triggers NETs formation. Moreover, *Bacteroides uniformis* or taurine inhibits the activation of NF-κB and IL-17 signaling pathways in neutrophils which harness NETs and alleviate pancreatic injury. Our findings establish roles of endogenous *Bacteroides uniformis*-derived metabolic and inflammatory products on suppressing NETs release, which provides potential insights of ameliorating HTGP through gut microbiota modulation.

Acute pancreatitis (AP) is caused by excessive pancreatic enzyme activation and pancreatic "self-digestion". The incidence is over 34 cases per 100,000 general population per year[1]. As a self-limiting mild disease, approximately 20–30% of AP patients develop to a severe form, which leads to critical illness and higher mortality rate[2]. Gallstone remains the leading cause of AP, followed by alcohol abuse[3]. Due to social determinants and diet shifts, the incidence of hypertriglyceridemic pancreatitis (HTGP) is increasing[4]. Accumulating evidence indicates that HTGP patients are complicated by infected pancreatic necrosis (IPN), organ failure, longer hospitalization, and

higher mortality[5,6]. Thus, understanding the mechanism of HTGP exacerbation will improve the therapeutics of HTGP.

Gut microbiota and tissue-resident bacteria are involved in pancreatic-specific disorders, including pancreatic cancer, diabetes and pancreatitis[7–9]. Variations in gut microbiota are associated with the severity and prognosis of AP patients[10]. Gut microbiota dysbiosis, intestinal barrier dysfunction, and translocation of enteric bacteria are considered as determining factors for IPN. Depletion of gut microbiota by broad-spectrum antibiotics pretreatment alleviates pancreatic injury in AP mouse models[10]. These findings indicate the crucial links

[1]Department of Pancreatic and Biliary Surgery, The First Affiliated Hospital of Harbin Medical University, Harbin 150001, China. [2]Key Laboratory of Hepatosplenic Surgery, Ministry of Education, Harbin 150001, China. [3]Department of Clinical Cancer Prevention, The University of Texas MD Anderson Cancer Center, Houston, TX, USA. [4]Department of Gastrointestinal Medical Oncology, The University of Texas MD Anderson Cancer Center, Houston, TX, USA. [5]Clinical Cancer Genetics Program, The University of Texas MD Anderson Cancer Center, Houston, TX, USA. [6]These authors contributed equally: Guanqun Li, Liwei Liu. ✉e-mail: lile@hrbmu.edu.cn; sunbei70@hrbmu.edu.cn

between "gut-pancreas axis" and the onset and progression of AP. Gut microbiota affects host physiology mainly by the production of metabolites and maintenance of immune homeostasis[11–13]. Gut microbiota-derived metabolites, including short-chain fatty acids and bile acids, have been described to activate adaptive immune responses that encompass diverse immunopathological outcomes in AP pathogenesis[14]. Recent study suggested that HTGP patients exhibit lower diversity of gut microbiota and increased abundance of potentially pathogenic bacteria, such as *Escherichia Shigella* and *Enterococcus*, which were associated with disease severity and poor prognosis in HTGP patients[15]. In addition, the imbalanced gut microbiota is associated with poor HTGP progression in HTGP models[16]. However, the impact of altered gut microbiota on disease severity, as well the potential mechanisms of bacteria affecting pancreatic injury in HTGP patients, remain to be fully understood.

In this study, we depicted the gut microbiota composition between HTGP patients, high triglycerides (HTG) controls and healthy individuals. The mechanism of HTGP-determined gut microbiota influencing local and systemic inflammation through bacterial metabolites and immune system talks was explored. Our findings address the importance of specific bacteria species on affecting pancreatic damage, and offer considerable strategies for the treatment of HTGP.

## Results

### Alternation of gut microbiota in HTGP patients

To explore the role of gut microbiota impacting pancreatic damage in HTGP patients, the clinical characteristics and gut microbiota composition between 25 HTGP patients, 16 healthy volunteers (HC) and 8 high triglycerides (HTG) controls were compared (Fig. 1a). The clinical features showed no difference among groups, beyond BMI and triglycerides level (Supplemental Table 1). We further analyzed the gut microbiota composition using 16 S ribosomal RNA (16 S rRNA) sequencing. Simpson index described a decreased alpha diversity in HTGP patients compared with healthy volunteers ($p < 0.05$) (Fig. 1b). Non-metric multidimensional scaling (NMDS) analysis showed clear separation of the microbial composition between two groups (Fig. 1c). The analysis of similarity (Anosim) indicated clear separation of the gut microbiota between HTGP patients and healthy volunteers (Anosim; R = 0.4928, P = 0.001). The gut microbiota community structures of the HTG controls and HTGP patients were also different (Anosim; R = 0.17, P < 0.05). These results indicate a less heterogeneous community structure in HTGP patients compared with both HTG and healthy controls. Next, we investigated the effects of HTGP on reshaping gut microbiota composition. At the phylum level, the decrease of *Firmicutes* and expansion of *Proteobacteria* were identified in HTGP patients compared with healthy volunteers (P = 0.003) (Supplementary Fig. 1a, b). HTGP patients showed higher abundances of *Enterococcaceae* and *Escherichia Shigella*, and lower abundances of *Bacteroides* and *Faecalibacterium* in the family and genus levels (Fig. 1d, Supplementary Fig. 1c, d). The differential bacterial species such as increased *Enterococcus faecium* and *E. coli* and decreased *Bacteroides uniformis* were found in HTGP patients (Fig. 1e). Linear discriminant analysis (LDA) of effect size (LEfSe) (LDA score > 4) suggested higher abundances of *Klebsiella*, *Enterococcus* and *Escherichia Shigella*, and lower beneficial bacteria, including *Faecalibacterium prausnitzii* and *Bacteroides uniformis* in HTGP patients (Fig. 1f). In particular, those bacteria were not identified as differential bacterial species between HC and HTG groups.

Next, the links between changes of gut microbiota and clinical characteristics that determine the outcomes of HTGP patients were investigated. The prevalence of systemic inflammatory response syndrome (SIRS) and ICU admission were negatively correlated with the abundances of *Faecalibacterium*, *Bacteroides uniformis* and *Collinsella aerofaciens*. Higher serum C-reactive protein (CRP) level was positively correlated with the abundances of *Subdoligranulum* and *Bifidobacterium*

(Fig. 1g, Supplementary Fig. 1e). These suggest that specific bacterial species are associated with disease severity and determine patients' prognosis.

### Gut microbiota dysbiosis exacerbates HTGP progression

To understand the role of imbalanced gut microbiota impacting pancreatic injury in HTGP, a fecal microbiota transplantation (FMT) mouse model was set up[17–19]. Mice were pretreated with a broad-spectrum antibiotics treatment (ABX) to deplete endogenous gut microbiota and then colonized with gut microbiota obtained from HTGP patients (FMT-HTGP), HTG controls (FMT-HTG) and healthy volunteers (FMT-HC) (Fig. 2a). ABX significantly reduced fecal bacteria load and decreased the alpha diversity of gut microbiota, suggesting that antibiotic treatment can successfully remove major gut resident microbiota prior to patients' fecal microbiota colonization (Supplementary Fig. 2a–c). In addition, antibiotic treatment had no effect on alleviating pancreas injury in HTGP mouse model (Supplementary Fig. 2d, e). Our results revealed that FMT-HTGP aggravated pancreatic injury and systemic inflammation, with increased pancreas histological score, serum amylase, lipase and IL-1β levels, compared with FMT-HC and FMT-HTG groups (Fig. 2b, c, Supplementary Fig. 6a, b). We also found lower diversity of gut microbiota and decreased abundances of *Bacteroides* and *Bacteroides uniformis* in HTGP patients compared with other types of AP patients (Supplementary Fig. 3a–d). Moreover, FMT-HTGP mice exhibited aggravated pancreatic damage compared with mice colonized with gut microbiota obtained from other types of AP patients (FMT-AP) (Supplementary Fig. 3e, f), suggesting that HTGP-associated changes in the gut microbiota are key factors for exacerbating pancreatic injury, and the unique gut microbial identities in HTGP may determine the different outcomes compared with other types of AP.

To explore whether patient-derived bacteria can affect HTGP development, the ability of colonization of human gut microbiota in mice was validated (Supplementary Fig. 2c, Supplementary Fig. 4a). We found that 54.0% and 59.4% of species (relative abundance > 0.001) identified from healthy volunteers and HTGP patients, respectively, were colonized in recipient mice (Supplementary Fig. 4b). These suggest that a significant proportion of microorganisms can be successfully transferred and colonized, which supports the utilization of FMT mouse model to understand the mechanism of gut microbiota modulating HTGP development.

Next, the gut bacterial taxonomic between FMT-HTGP and FMT-HC mice was compared, and FMT-HTGP mice exhibited decreased alpha diversity (Fig. 2d). Ordination of Bray-Curtis dissimilarity by principal coordinate analysis (PCoA) revealed significant divergence between FMT-HC and FMT-HTGP mice (Anosim; R = 0.4244, P = 0.002) (Fig. 2e). At the genus level, the abundances of several potentially facultative pathogens such as *Enterococcus* and *Escherichia Shigella* were increased, whereas *Akkermansia* and *Bacteroides* were decreased in FMT-HTGP mice (Supplementary Fig. 4c). At the species level, the potentially beneficial bacteria, *Bacteroides uniformis*, was reduced in FMT-HTGP group (Fig. 2f, Supplementary Fig. 4d). LEfSe revealed that *Bacteroides*, including *Bacteroides massiliensis*, *Bacteroides uniformis* and *Bacteroides acidifaciens*, were relatively increased, while *Lactobacillus* and *Vagococcus* were decreased in FMT-HC mice (Fig. 2g). Our data suggest that altered gut microbiota plays causal effect on pancreatic injury and systemic inflammatory response in HTGP.

### Gut microbiota dampens HTGP progression by the reduction of neutrophil extracellular traps

The interplays between gut microbiota and host immune system have been described. The immune regulatory roles of gut microbiota were further explored in FMT mouse models and increased proportion of proinflammatory neutrophils and macrophages in pancreatic tissues were found in FMT-HTGP mice compared with FMT-HC mice (Fig. 3a,

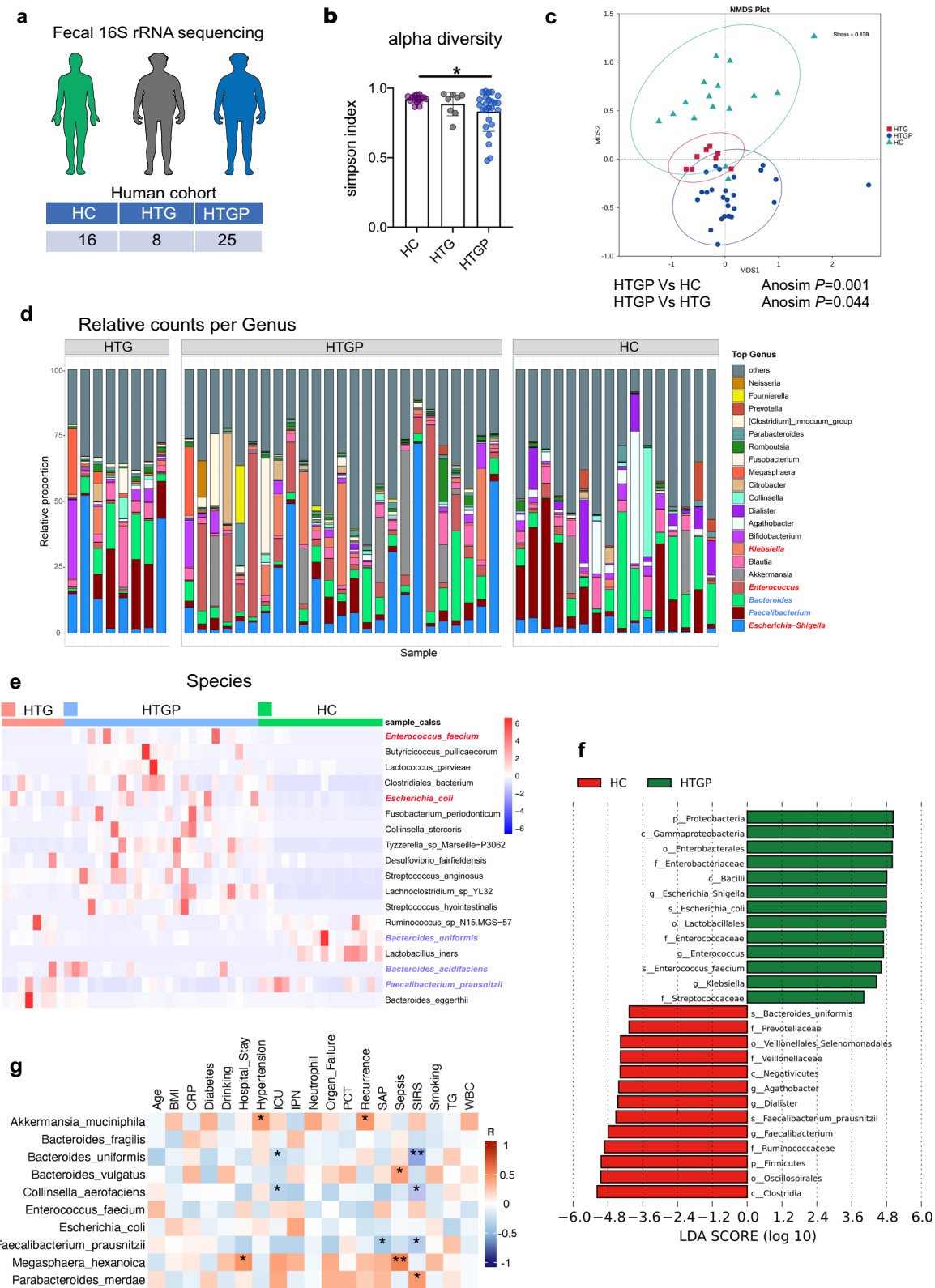

**Fig. 1 | Alternation of gut microbiota in HTGP patients. a** Schematic representation of human study design (healthy volunteers, $n = 16$; high triglycerides matched controls, $n = 8$; HTGP patients, $n = 25$). **b** Alpha diversity (based on simpson, $P = 0.0421$). **c** Nonmetric multidimensional scaling (NMDS) plot clustering differential microbial distributions between groups. **d** Relative abundance of gut bacteria in the genus level. Top 20 average abundance of microbes were compared. **e** The hierarchical clustering heatmap of differential bacterial species (*Bacteroides uniformis*, HC vs HTGP, $P = 0.0001$). **f** LEfSe indicating differential bacterial taxa between HC and HTGP groups. LDA score of differentially abundant bacterial taxa (LDA > 4, p: Phylum, c: Class, o- Order, f: Family, g: Genus, s: Species). **g** Correlation analysis between different genera and clinical outcomes in HTGP patients, as well as indicators of disease severity. *P* values were determined by two-tailed ordinary one-way ANOVA with the Tukey post hoc test or Student's *t*-test. Data was represented as mean ± SEM. *$p < 0.05$, **$p < 0.01$, ***$p < 0.001$. Source data are provided as a Source data file.

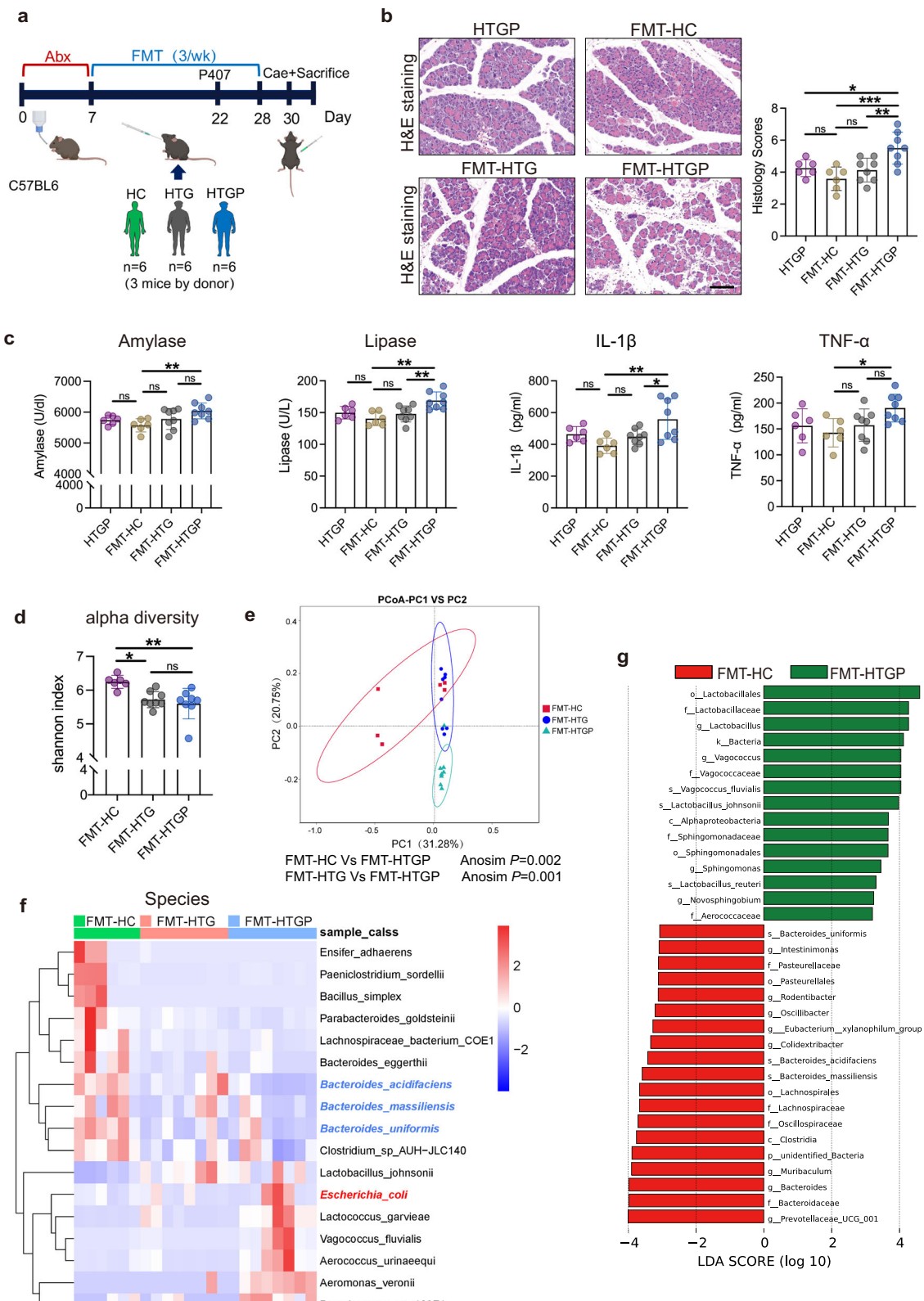

Supplementary Fig. 5a). Tissue transcriptome sequencing was performed to explore expression patterns of genes between two groups from three biological replicates (Supplementary Table 3), and gene enrichment analysis indicated that inflammatory responses such as leukocyte activation, neutrophil infiltration and neutrophil degranulation were increased in FMT-HTGP mice compared with FMT-HC mice (Fig. 3b, Supplementary Fig. 5b). Increased neutrophil infiltration by

FMT-HTGP was confirmed by IHC staining (Fig. 3c and Supplementary Fig. 5c, d). Gene set enrichment analysis (GSEA) revealed upregulated neutrophil extracellular traps formation in FMT-HTGP mice which is considered as a determinate pro-inflammatory biological process (Fig. 3d). To further assess the heterogeneity of HTGP driven by different gut microbiota, single-cell RNA sequencing (scRNA-seq) was conducted in pancreatic tissues (Fig. 3e). After quality control of raw

**Fig. 2 | Gut microbiota dysbiosis exacerbates HTGP progression. a** FMT experimental design, C57BL/6 mice were colonized with gut microbiota obtained from HC, HTG and HTGP donors (n = 6). FMT fecal microbiota transplantation, HTGP hypertriglyceridemic pancreatitis, HTG high triglycerides matched controls, HC healthy volunteers. **b** Representative images of pancreas by H&E staining and quantification of histology score (Scale bar = 100 μm, HTGP, n = 6; FMT-HC, n = 6; FMT-HTG, n = 8; FMT-HTGP, n = 8) (FMT-HC vs FMT-HTGP, P = 0.0008, FMT-HTG vs FMT-HTGP, P = 0.0097). **c** Serum amylase, lipase, IL-1β and TNF-α levels (HTGP, n = 6; FMT-HC, n = 6; FMT-HTG, n = 8; FMT-HTGP, n = 8) (Amylase: FMT-HC vs FMT-HTGP, P = 0.0027; lipase: FMT-HC vs FMT-HTGP, P = 0.001, FMT-HTG vs FMT-HTGP, P = 0.0092; IL-1β: FMT-HC vs FMT-HTGP, P = 0.0039, FMT-HTG vs FMT-HTGP, P = 0.0494; TNF-α: FMT-HC vs FMT-HTGP, P = 0.0307). **d** Alpha diversity, as revealed by shannon index (FMT-HC, n = 6; FMT-HTG, n = 8; FMT-HTGP, n = 8) (FMT-HC vs FMT-HTGP, P = 0.0044, FMT-HC vs FMT-HTG, P = 0.0199). **e** Principal coordinate analysis (PCoA) analysis showing beta diversity by Bray-Curtis metric distance. **f** The hierarchical clustering heatmap of differential bacterial species among FMT-HC, FMT-HTG and FMT-HTGP groups (FMT-HC, n = 6; FMT-HTG, n = 8; FMT-HTGP, n = 8). **g** LEfSe indicating differential bacterial taxa between FMT-HC and FMT-HTGP groups. LDA score of differentially abundant bacterial taxa (p: Phylum, c: Class, o- Order, f: Family, g: Genus, s: Species). One-way ANOVA with the Tukey post hoc test was performed to do multiple comparisons test. Data was represented as mean ± SEM. *p < 0.05, **p < 0.01, ***p < 0.001. Source data are provided as a Source data file.

data, a total of 29,082 cells were obtained for subsequent analysis (Supplementary table 4). The markers of each cell type were shown (Supplementary Fig. 5g), and the proportion of neutrophils was increased in FMT-HTGP group compared with FMT-HC group (Fig. 3f, Supplementary Fig. 5e, f). Correspondingly, the formation of neutrophil extracellular trap pathway was activated in the neutrophils of FMT-HTGP mice (Fig. 3g, h). The IL-17 and NF-κB signaling pathways that can drive NETs formation were also activated in FMT-HTGP mice (Supplementary Fig. 5h)[20,21]. In agreement with the pathway analysis, FMT-HTGP increased the expressions of *Padi4*, *Nfkb* and *Il17a* in the pancreas that are known as key regulators of NETs formation (Fig. 3i). Briefly, HTGP-derived gut microbiota accelerates excessive inflammatory immune responses through neutrophil infiltration and NETs formation.

Gut microbiota disrupts intestinal barrier function and immune system that cause AP progression[22]. Our findings revealed that HTGP disrupted colonic barrier in FMT-HTGP mice, whereas had no effect on intestinal barrier injury compared with FMT-HC mice (Supplementary Fig. 6c, d). These suggest that HTGP-improved gut permeability allows bacteria and produced metabolites to enter circulation that are involved in pancreatic disorders. Notably, we found that FMT-HTGP increased the proportions of IL-17A-producing CD4+ lymphocytes (p < 0.001) and improved the Th17/Treg ratio in lamina propria (Supplementary Fig. 6e). The released IL-17A may translocate into the pancreas and accelerate NETs formation. Neutrophils generate extracellular reticular structures composed of histones, granule proteins and depolymerized genomic DNA through a unique form of cell death called "NETosis"[23,24]. Neutrophils-released CitH3 can be used for measuring NETs, we found a marked reduction of CitH3 and Mpo co-localization in FMT-HC mice, indicating that "healthy" gut microbiota restrained NETs formation (Fig. 3k). We also found higher propensity for NETs, illustrated by numerous large, intricately connected NETs (Fig. 3j), as well as increased serum levels of CitH3 and dsDNA in FMT-HTGP mice (Fig. 3l, Supplementary Fig. 5i). These suggest that HTGP-derived gut microbiota exacerbates pancreatic injury through affecting NETs formation.

### *Bacteroides uniformis* alleviates HTGP through the inhibition of neutrophil infiltration and reduction of NETs

To identify which bacteria species influence NETs formation in HTGP, the gut microbiota composition between HTGP patients and mouse models was compared. Three candidate bacteria, *Bacteroides uniformis*, *Bacteroides acidifaciens* and *Lactococcus garvieae*, were found significantly decreased in FMT-HTGP mice compared to FMT-HC mice (p < 0.05) (Supplementary Fig. 7a, b). A lower abundance of *Bacteroides uniformis* was occurred in both HTGP patients and recipient mice (Fig. 4a). Moreover, the abundance of *Bacteroides uniformis* was found to be negatively correlated with neutrophils recruitment that was considered as a candidate protector for HTGP (Supplementary Fig. 7c). Consistent with these findings, the upregulated abundance of *Bacteroides uniformis* was negatively associated with the occurrence of HTGP (Supplementary Fig. 7d). To understand whether *Bacteroides uniformis*

dampens pancreatic injury through limiting neutrophils recruitment, mice were orally gavaged with *Bacteroides uniformis* strain and then HTGP mouse model was generated (Fig. 4b). We observed that *Bacteroides uniformis* restrains pancreatic damage by the reduction of neutrophils infiltration and NETs (Fig. 4c–e, Supplementary Fig. 7e–g).

Glycosylphosphatidylinositol-anchored high-density lipoprotein-binding protein 1 (GPIHBP1) deficiency mice exhibit higher triglyceride (TG) level and severe chylomicronemia that have been recommended for building spontaneous HTGP model[25]. We found that administration of *Bacteroides uniformis* alleviated pancreatic injury in GPIHBP1−/− mice (Fig. 4f–h). We then explored whether *Bacteroides uniformis* alleviates pancreatic injury through modulating neutrophils infiltration, and found that both Ly6g neutralization and *Bacteroides uniformis* administration can ameliorate pancreatic injury, while the combination of neutrophils depletion and *Bacteroides uniformis* administration exhibited no amplification of anti-inflammatory roles (Supplementary Fig. 8a–d). These demonstrate that *Bacteroides uniformis* alleviates pancreatic injury through neutrophils-dependent manner. Consequently, *Bacteroides uniformis* reduced web-like structures in the inflamed pancreas that restrained NETs formation (Fig. 4i–k). Taken together, our results demonstrate that *Bacteroides uniformis* ameliorates HTGP through limiting neutrophil infiltration and reducing NETs.

### *Bacteroides uniformis*-produced taurine ameliorates HTGP

Gut microbiota-derived metabolites are recognized as central regulators in inflammatory disorders. We then profiled the serum metabolites in FMT mouse models and found 29 differential metabolites between FMT-HC and FMT-HTGP mice, (Fig. 5a) (VIP > 1.5, p < 0.05). Taurine metabolism, bile acid biosynthesis and nitrogen metabolism pathways were top predictive pathways modulated by FMT-HC (Fig. 5b, Supplementary Fig. 9a). Next, the correlation between significantly different metabolites and gut microbial signatures were explored, and taurine level was positively correlated with the abundances of *Bacteroides* and *Bacteroides uniformis* (Fig. 5c, Supplementary Fig. 9b, c). Furthermore, we found that the level of taurine was significantly elevated in the fecal and serum samples after *Bacteroides uniformis* administration, suggesting that *Bacteroides uniformis*-associated taurine production may alleviate pancreatic injury (Fig. 5d, e).

We then explored whether administration of taurine can attenuate pancreatic damage and systemic inflammation in HTGP (Fig. 5f), and found that taurine reduced serum proinflammatory cytokines and ameliorated pancreatic injury through suppressing neutrophils infiltration and reducing NETs in an inducible HTGP model[25,26] (Fig. 5g–j). We then investigated the effects of taurine on alleviating HTGP in GPIHBP1−/− mice, and found that oral supplementation of taurine elevated its levels in the pancreas and circulation that reduced NETs formation and alleviated pancreatic injury and systemic inflammation (Fig. 5k–n, Supplementary Fig. 9d–f, Supplementary Fig. 10a–e). In addition, taurine reduced IL-17A level in the pancreas that inhibited NETs release (Supplementary Fig. 9g). These suggest that *Bacteroides uniformis*-produced taurine restrains HTGP development in regardless of the factors of hypertriglyceridemia.

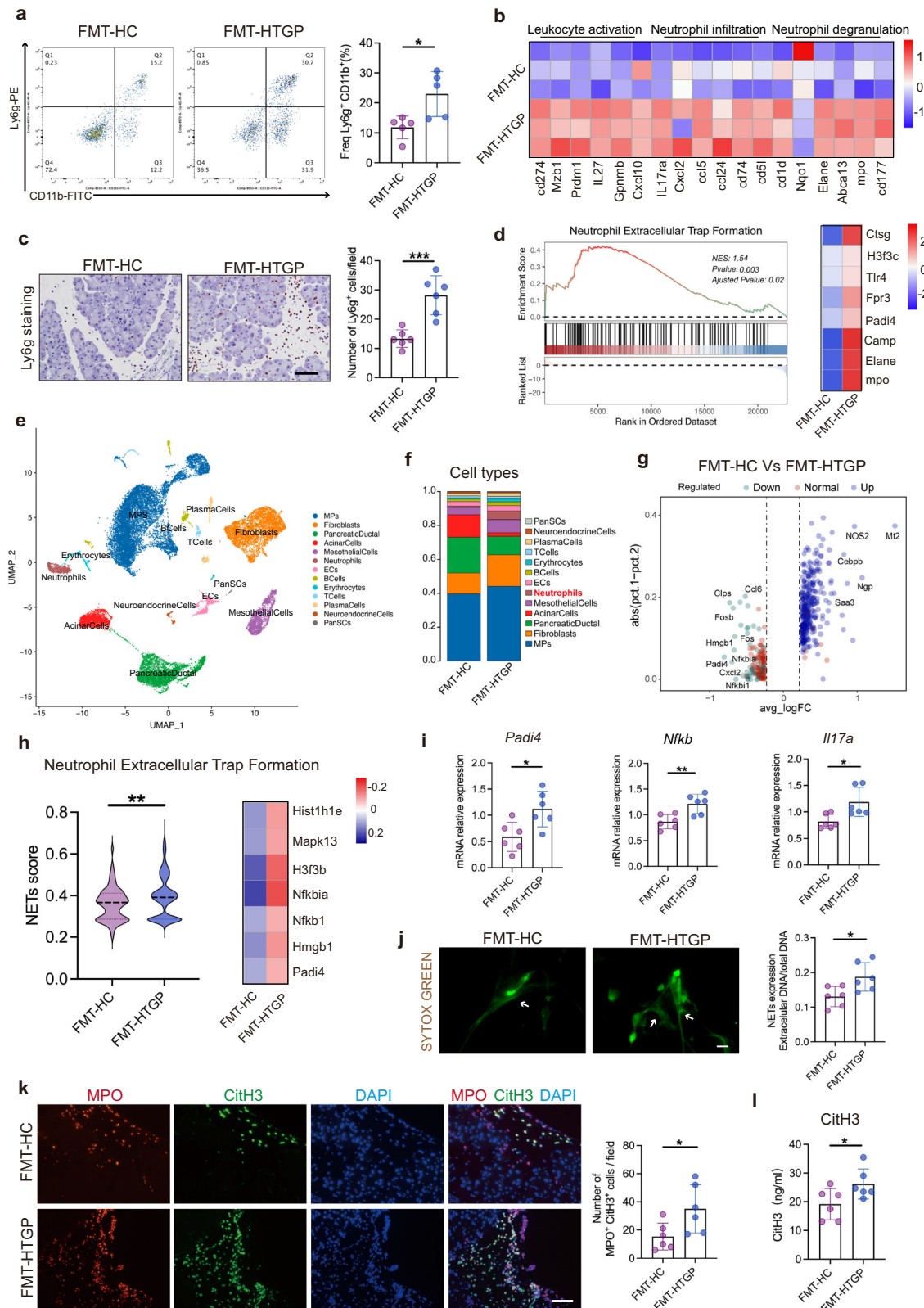

## Taurine alleviates HTGP through NETs inhibition

Peptidyl arginine deiminase 4 (PAD4) is particularly expressed in neutrophils that drives NETs formation. PAD4$^{-/-}$ mice were used to validate the hypothesis of taurine ameliorating HTGP through suppressing NETs (Fig. 6a)[27]. PAD4$^{-/-}$ mice exhibited mild pancreatic injury and lesser neutrophils infiltration, and supplementation of taurine failed to alleviate HTGP (Fig. 6b–e). Next, we observed the equivalent

of diminished Mpo/CitH3 double staining-positive and similar Sytox Green positive area in the PAD4$^{-/-}$ mice with or without taurine treatment, suggesting that taurine alleviates HTGP through NETs-specific mechanism (Fig. 6f, g, i). To determine whether taurine attenuates pancreatic injury by restraining NETs in vitro, isolated neutrophils were stimulated by 12 phorbol 13-myristate acetate (PMA) (100 nM), a potent NETs inducer, and supernatant cell-free DNA was then

**Fig. 3 | Gut microbiota dampens HTGP progression by the reduction of neutrophil extracellular traps. a** Flow cytometric quantification and statistical analysis of neutrophils infiltration in the pancreas between FMT-HC and FMT-HTGP groups (*n* = 5, *P* = 0.018). **b** Different genes enriched in leukocyte activation, neutrophil infiltration and neutrophil degranulation pathways between FMT-HC and FMT-HTGP mice (FMT-HC, *n* = 3; FMT-HTGP, *n* = 3). **c** Representative images and quantification of Ly6g positive cells by IHC (Scale bar = 50 μm; *n* = 6, *P* = 0.0006). **d** GSEA snapshots of neutrophil extracellular traps formation pathway enrichment analysis and gene signatures enriched in neutrophil extracellular traps formation pathway between FMT-HC and FMT-HTGP mice. **e** A UMAP plot displaying distinct clusters of 29,082 cells obtained from FMT-HC and FMT-HTGP mice (*n* = 3 biologically independent animals per group). **f** Proportion of cell types. **g** Volcano plot showing differentially expressed genes in neutrophils between FMT-HC and FMT-HTGP mice. FC fold change. **h** Enrichment score of neutrophil extracellular traps formation pathway in the neutrophils between FMT-HC and FMT-HTGP mice, and identification of gene signatures associated with NETs formation (*Padi4, Hmgb1, Nfkb1, Nfkbia, H3f3b, Mapk13* and *Hist1h1e, P* = 0.0032). **i** Quantitative RT-PCR tested expressions of *Padi4, Nfkb*, and *Il17a* in pancreas between FMT-HC and FMT-HTGP mice (*n* = 6) (*Padi4: P* = 0.0143; *Nfkb: P* = 0.0049; *Il17a: P* = 0.0148). **j** Representative immunofluorescence staining images of SYTOX to map NETs in neutrophils isolated from FMT-HC and FMT-HTGP mice. Green represents extracellular (ex) DNA, as stained by SYTOX Green (Scale bar = 25 μm, *n* = 6, *P* = 0.0204). **k** Representative immunofluorescence images and quantification of Mpo⁺CitH3⁺cells in the pancreas (Scale bar = 50 μm, *n* = 6, *P* = 0.0336). Green, CitH3 positive cell; Red, Mpo positive cell. **l** Serum CitH3 level by ELISA (*n* = 6, *P* = 0.0456). For intergroup comparison of unpaired data, two-tailed Student's *t*-test was used for normal distribution and Mann–Whitney test was used for nonparametric data. Data was represented as mean ± SEM. \**p* < 0.05, \*\**p* < 0.01, \*\*\**p* < 0.001. Source data are provided as a Source data file.

measured. Our data showed that taurine reduced NETs in response to PMA stimulation and decreased DNA level in the cell-free supernatant (Fig. 6h). These uncover the predominant role of targeting NETs by taurine supplementation that confines HTGP development.

## Taurine suppresses NETs and restrains inflammatory response in HTGP patients

To understand the links between *Bacteroides uniformis*-produced taurine and the severity of HTGP in patients, the association between the abundance of *Bacteroides uniformis* and serum taurine level was plotted, and the correlation between NETs formation and disease severity was explored (Fig. 7a). Our results revealed that severe HTGP patients exhibited increased NETs and decreased taurine level in the serum (Fig. 7b). Meanwhile, the level of taurine was positively associated with the abundance of *Bacteroides uniformis*, and was negatively correlated with markers of NETs (Fig. 7c, d). To dissect whether taurine can inhibit NETs in vitro, neutrophils were isolated from HTGP patients and healthy volunteers, and the levels of NETs were examined. We found increased NETs level in HTGP patients compared with healthy volunteers, and taurine markedly confined the release of NETs (Fig. 7e). Moreover, neutrophils isolated from mice were incubated with serum from different types of donors, we found that serum obtained from HTGP patients indicated higher capability of inducing the release of NETs compared with healthy volunteers (Fig. 7f). These results highlight the negative relevance between *B.uniformis*-produced taurine and NETs formation in clinic.

To understand the cellular mechanisms of *B.uniformis*-derived taurine affecting NETs formation, neutrophils were isolated from peripheral blood of healthy individuals and treated with PMA to induce NETs (Supplementary Fig. 10f). The mRNA levels of *Nfkb* and *Il17a* as well as the protein expressions of NF-κB signaling pathway were tested. Our results indicated that taurine reduced the gene expressions of *Nfkb* (*P* < 0.01) and *Il17a* (*P* < 0.05) (Fig. 7h, i), as well as the protein levels of NF-κB and its phosphorylation (*P* < 0.05) (Fig. 7g). These suggest that *B. uniformis*-derived taurine limits the activations of IL-17 and NF-κB signaling pathways in the neutrophils that repress NETs and pancreatic injury in HTGP.

## Discussion

As the second leading cause of AP, hypertriglyceridemia accounts for 20–30% of all cases. HTGP occurs in younger age, with higher complication rates and mortality, the pathogenesis remains unknown[28,29]. Despite that previous studies have shown that gut microbiota dysbiosis affects HTGP development, the mechanism of bacteria influencing pancreatic injury and systemic inflammation remains unclear[15]. In this study, the associations between changes of gut microbiota and the prognosis of HTGP patients have been identified. HTGP patients exhibited increased abundances of *Escherichia Shigella* and *Enterococcus*, and decreased abundances of *Bacteroides* and *Faecalibacterium*, and the downregulated abundances of *Faecalibacterium prausnitzii* and *Bacteroides uniformis* indicated severe complications and poor outcome. Herein, we demonstrate that gut microbiota can be holistic approach for the diagnostics and severity evaluation in HTGP, and decipher the aspects of microbiota-modulated immune homeostasis and pancreatic pathologies in clinical perspectives.

Depletion of gut microbiota alleviates caerulein-induced AP[10], but not HTGP, establishing diverse aspects of pro-inflammatory roles induced by gut microbiota in pancreatic inflammatory disorders[30,31]. FMT-HTGP exacerbated pancreatic damage and systemic inflammation compared with FMT-HC, FMT-HTG, even FMT-AP, suggesting that the imbalanced gut microbiota caused by hyperglycemia-associated pancreatic damage, rather than metabolic disruptions or other aetiological factors of pancreatic injury, determines the deterioration of HTGP. Gut microbiota-immune interactions are regarded as pivotal risk factors of AP evolvement[14,32,33]. We uncovered that gut microbiota, particular *Bacteroides uniformis*, eliminated pancreatic injury and systemic inflammatory responses in HTGP by restraining neutrophil infiltration. Gut microbiota alters host metabolism and influences immune homeostasis which affect AP progression[34–36]. *Bacteroides uniformis* was identified as a beneficial commensal for eliminating HTGP progression. *Bacteroides uniformis* and its metabolites are able to ameliorate obesity and liver steatosis[37,38]. In response to a pathogenic bacterial stimulus, *Bacteroides uniformis* manipulates T regulatory cells and B cells activation, as well as induces cytokine production and release in macrophages and dendritic cells[39]. Here, we found that FMT-HC or colonization by *Bacteroides uniformis* improved the production of taurine in the gut, as well as the systemic level of taurine. Taurine is involved in the maintenance of neutrophil redox homeostasis and restrains inflammatory dissemination by controlling cytokine release[40,41]. Gut microbiota-produced taurine can shield host-microbiota interface that potentiates resistance to pathogens infection[42]. Due to pancreatitis-caused intestinal inflammation and gut barrier dysfunction, taurine can easily enter the circulation and travel to the pancreas that contributes to anti-inflammatory effects. Our results indicate a crucial interplay between gut microbial metabolite and immunomodulatory effects in HTGP development.

In this study, we uncovered that gut microbiota manipulated neutrophils-associated genes such as *Padi4* and *Il17a* that affected NETs, which is considered as the main mechanism of influencing local and systemic immune responses in HTGP. In spite of roles in host defense, aberrant and prolonged release of NETs also contribute to pro-inflammatory processes[43,44]. NETs drive trypsin activation and cause pancreatic duct obstruction that exacerbate pancreatic damage in AP[45]. Pioneering studies provide insights into NETs from a microecological perspective[46]. NETs can trap or kill pathogenic bacteria, in turn, bacteria can also trigger or inhibit NETs formation. Our results revealed that *Bacteroides uniformis*-derived taurine suppressed NETs in both pancreas and peripheral circulation. Supplementation of

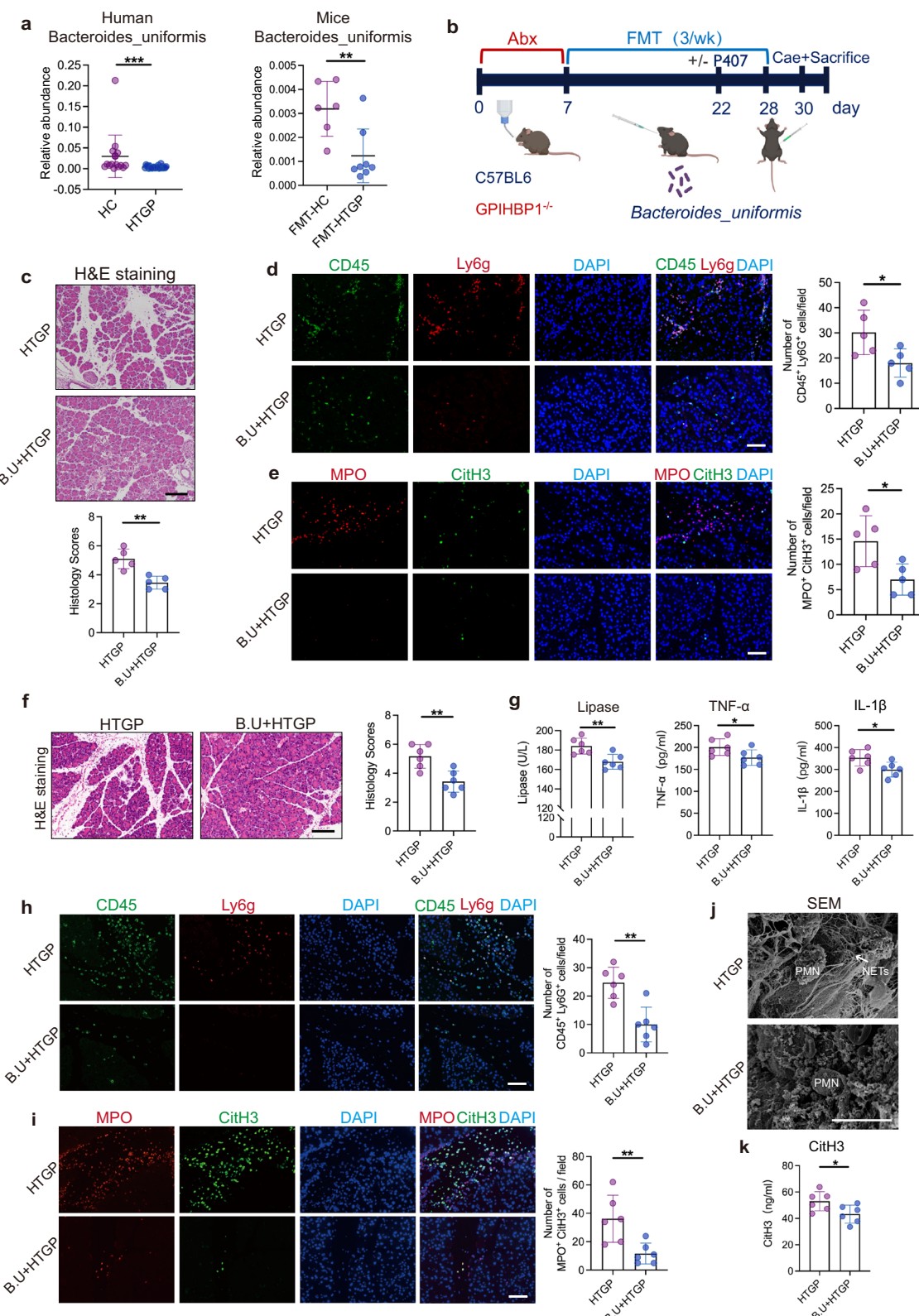

taurine failed to ameliorate pancreatic injury in PAD4[−/−] mice, indicating the direct interaction between taurine-dependent anti-inflammatory effects and reduction of NETs. Our study provides new insights into gut microbiota and its metabolic products involving in NETs-mediated immune response.

The mechanism of gut microbiota on NETs manipulation needs to be elucidated[47]. Taurine confines both PMA and hypochlorous acid-induced NETs release, and rescues neutrophils from programmed cell death[48]. In addition, taurine deactivates MAPK signaling pathway and disrupts NADPH oxidase that reduce NETs formation[49]. We observed that FMT-HTGP activates IL-17 and NF-κB signaling pathways within neutrophils, and the increased levels of IL-17 and phosphorylation of NF-κB can be blocked by taurine supplementation. These suggest the metabolic aspects of gut microbiota-induced neutrophils recruitment and NETs release in damaged pancreatic tissue. IL-17 is a potent inducer of NETosis, and blockade of IL-17 reduces NETs formation[50].

**Fig. 4 | *Bacteroides uniformis* alleviates HTGP through inhibition of neutrophils and reduction of NETs. a** The abundance of *Bacteroides uniformis* in human fecal samples and gut microbiota of recipient mice (HC, n = 16, HTGP patients, n = 25; FMT-HC, n = 6, FMT-HTGP, n = 8) (HC vs HTGP, P = 0.0001; FMT-HC vs FMT-HTGP, P = 0.0074). **b** Schematic representation of *Bacteroides uniformis* supplementation design. P407-induced HTG mice were randomly divided into Control and B.U groups (n = 5 per group) for generating HTGP models. GPIHBP1⁻/⁻ mice were randomly divided into Control and B.U groups (n = 6 per group) for generating HTGP models. **c** Representative images of H&E staining and quantification of histology score in P407-induced HTGP model (Scale bar = 100 μm, n = 5 biologically independent animals per group, P = 0.0019). **d** Representative immunofluorescence images of CD45⁺ Ly6g⁺ cells in the pancreas (Scale bar = 50 μm, n = 5, P = 0.0311). **e** Representative immunofluorescence images and quantification of Mpo⁺ CitH3⁺ cells in the pancreas (Scale bar = 50 μm, n = 5, P = 0.0205). **f** Representative images

of H&E staining and quantification of histology score in GPIHBP1⁻/⁻ mice (Scale bar=100 μm, n = 6 biologically independent animals per group, P = 0.003). **g** Serum lipase, TNF-α and IL-1β levels (n = 6 biologically independent animals per group) (Lipase: P = 0.0052; TNF-α: P = 0.048; IL-1β: P = 0.0255). **h** Representative immunofluorescence images and quantification of CD45⁺Ly6g⁺ cells in the pancreas (Scale bar = 50 μm, n = 6, P = 0.0014). **i** Representative immunofluorescence images and quantification of Mpo⁺ CitH3⁺ cells in the pancreas (Scale bar = 50 μm, n = 6, P = 0.0079). **j** Scanning electron microscopy showing extracellular web-like structures in the pancreas (Scale bar = 10 μm). **k** Serum CitH3 by ELISA (n = 6 biologically independent animals per group, P = 0.0378). For intergroup comparison of unpaired data, two-tailed Student's t-test was used for normal distribution and the Mann−Whitney test was used for nonparametric data. Data was represented as mean ± SEM. *p < 0.05, **p < 0.01, ***p < 0.001. Source data are provided as a Source data file.

Gut microbiota maintains epithelial barrier integrity and shapes mucosal immune system. Our findings demonstrated that lack of bacteria-produced taurine impaired the balance of Th17/Treg cells and enhanced IL-17A release from gut, which is considered as secondary mechanism of NETs induction in HTGP development.

Here, we describe an entirely new approach to manage HTGP-associated inflammatory processes through alteration of gut microbiota. Our study extends understandings of how gut microbiota-modulated metabolic product regulates host immunity in pancreatic inflammatory disorder, and provides profound evidence of designing *Bacteroides uniformis*-based probiotics or taurine-associated dietary intervention for the treatment of HTGP.

## Methods

### Sample collection
This study was approved by the Ethics Committee of the First Affiliated Hospital of Harbin Medical University (IRB-AF/SC-05/05.0), and all participants signed the informed consent. The inclusion criteria included a diagnosis of AP according to the 2012 revised Atlanta criteria[51]. HTGP was defined as serum triglycerides >1000 mg/dL or the serum TG levels between 500 to 1000 mg/dL accompanied by chylous fasting serum without other etiologies of AP. Patients must be presented in the hospital within 24 h after disease onset. The major exclusion criteria included chronic pancreatitis, inflammatory bowel disease, cancer, irritable bowel syndrome, gastroenteritis, and use of antibiotics, probiotics or laxatives within two months. The fecal samples were collected from 25 HTGP patients, eight serum triglycerides-matched controls, 16 healthy volunteers and 20 AP patients. Age and gender matched healthy volunteers were recruited. The clinical characteristics showed no difference between HTGP patients and healthy volunteers, except BMI and levels of triglycerides levels (Supplementary Table 1). The fecal samples were collected on the first day of hospital admission. Fecal samples were divided into aliquots and stored at −80 °C for further analysis.

### 16 S ribosomal RNA gene amplicon sequencing
Total bacterial DNA was extracted from fecal samples using the Power Soil DNA Isolation Kit (MO BIO Laboratories) according to the manufacturer's protocol, and the quality and quantity of DNA were further assessed. The V3-V4 region of the bacterial 16 S rRNA gene was amplified with the common primer pair (Forward primer, 5′-ACTCC-TACGGGAGGCAGCA-3′; reverse primer, 5′-GGACTACHVGGGTWTC-TAAT-3′) combined with adapter sequences and barcode sequences. All PCR mixtures contained 15 μL of Phusion® High-Fidelity PCR Master Mix (New England Biolabs), 0.2 μM of each primer and 10 ng target DNA, and cycling conditions consisted of a first denaturation step at 98 °C for 1 min, followed by 30 cycles at 98 °C (10 s), 50 °C (30 s) and 72 °C (30 s) and a final 5 min extension at 72 °C.

An equal volume of 1× loading buffer (contained SYBR green) was mixed with PCR products and performed electrophoresis on 2% agarose gel. PCR products were mixed in equal proportions, and

Qiagen Gel Extraction Kit (Qiagen, Germany) was used to purify mixed PCR products. The sequencing libraries were generated with NEBNext® Ultra™ IIDNA Library Prep Kit (Cat No. E7645) and sequenced on Illumina NovaSeq platform and 250 bp paired-end reads were generated.

### ASVs denoise and species annotation
For the effective tags obtained previously, denoise was performed with DADA2 or deblur module in the QIIME2 software (Version QIIME2-202006) to obtain initial ASVs (Amplicon Sequence Variants), and then ASVs with abundance less than 5 were filtered out. Species annotation was performed using QIIME2 software. In order to explore phylogenetic relationship of each ASV and the differences of the dominant species between samples, multiple sequence alignment was performed using QIIME2 software. The absolute abundance of ASVs was normalized using a standard of sequence number corresponding to the sample with the least sequences. Subsequent analysis of alpha diversity and beta diversity were performed based on the output normalized data.

### Alpha diversity
Alpha diversity was estimated by calculating the Chao1 richness, observed_species, Shannon-Wiener diversity index, and Simpson diversity index from different perspectives, which were focused on the microbial community abundance and evenness.

### Beta diversity
Beta diversity (between-sample diversity) was estimated by the Bray-Curtis and Weighted_unifrac dissimilarity index matrices using the Anosim method in the vegan R package. Principal Coordinate Analysis (PCoA) was performed to obtain principal coordinates and visualize differences of samples in complex multi-dimensional data. Two-dimensional PCoA results were displayed using ade4 package and ggplot2 package in R software (Version 2.15.3). Non-metric multi-dimensional scaling (NMDS), an indirect gradient analysis, was performed in a 2-dimensional con- figuration ('R' vegan function 'metaMDS'). The functions of adonis and anosim were used to explore the differences in community structure between groups. MetaStat and t-test analysis were performed to figure out different species at each taxonomic levels (Phylum, Class, Order, Family, Genus, Species). The predominance of bacterial communities between groups was analyzed using the linear discriminant analysis (LDA) effect size (LEfSe) (LDA score (log10) = 4 as cutoff value).

### Animals and experimental model
C57BL/6 mice (male, 6−8 weeks old) were purchased from Cyagen Biosciences Inc. (Suzhou, China). Two types of HTG (poloxamer 407 (P407)−induced and GPIHBP1⁻/⁻) models were employed. GPIHBP1⁻/⁻ mice and PAD4⁻/⁻ mice, were purchased from Cyagen Biosciences Inc. Age- and sex-matched wild-type C57BL/6 N mice were used as controls. All animals were housed in specific pathogen-free environment with controlled conditions, including a constant temperature (22 ± 2 °C), 12-

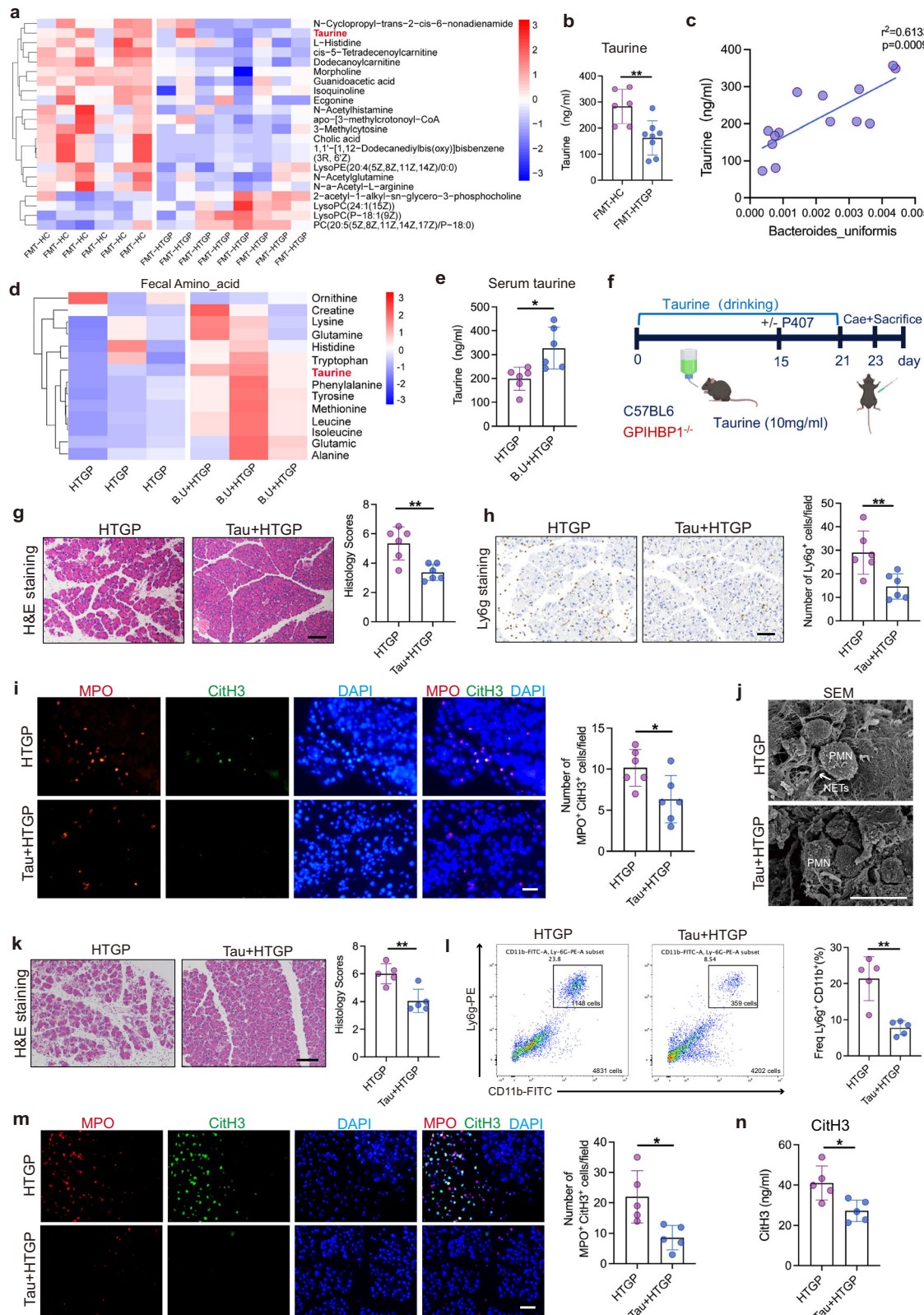

h light/dark cycle and 50 ± 5% humidity. The animals had free access to water and standard chow (Xie Tong Biotechnology, SFS9112). The sterile drinking water was replaced twice a week. The above animals were commercially available and acclimated for at one week before experiment started. All experimental procedures involving animals were carried out according to protocols approved by the Institutional Animal Ethics Committee of The First Affiliated Hospital of Harbin Medical University.

For the inducible HTG model, P-407 (Poloxamer 407, Sigma-Aldrich Co., USA) was administered intraperitoneally to mice every other day (0.5 g/kg body weight (b.w)) for seven consecutive days, the control mice were injected with PBS[25,26]. Twelve-week-old GPIHBP1[−/−] mice were considered with stable serum HTG level, and were used for generating HTGP model[52,53]. Caerulein (CER)-AP was performed by 10 hourly intraperitoneal injections of caerulein (50 µg/kg, Sigma-Aldrich Co., USA); and control mice were received

**Fig. 5 | *Bacteroides uniformis*-derived taurine ameliorates HTGP. a** Heatmap plotting different serum metabolites between FMT-HC and FMT-HTGP mice (FMT-HC, $n = 6$; FMT-HTGP, $n = 8$). **b** The levels of serum taurine tested by ELISA (FMT-HC, $n = 6$, FMT-HTGP, $n = 8$, $P = 0.0054$). **c** Correlation between the abundance of *Bacteroides uniformis* and serum taurine level was analyzed using Spearman's correlation analysis. **d** Heatmap showing differential fecal metabolites between HTGP and B.U + HTGP groups ($n = 3$). **e** Serum taurine levels between HTGP and B.U + HTGP groups ($n = 6$, $P = 0.0104$). **f** Scheme for taurine treatment. P407-induced HTG mice were randomly divided into Control and Tau-treated groups ($n = 6$ per group). GPIHBP1$^{-/-}$ mice were randomly divided into two Control and Tau-treated groups ($n = 5$ per group). **g** Representative images of H&E staining and quantification of histology score in P407-induced HTGP model (Scale bar = 100 μm, $n = 6$ biologically independent animals per group, $P = 0.0033$). **h** Representative images and quantification of Ly6g positive cells by IHC (Scale bar = 50 μm, $n = 6$ biologically independent animals per group, $P = 0.008$). **i** Representative immunofluorescence images of CitH3$^+$ Mpo$^+$ cells in the pancreas (Scale bar = 50 μm, $n = 6$, $P = 0.0274$). **j** Scanning electron microscopy showing extracellular web-like structures in the pancreas (Scale bar = 10 μm). **k** Representative images of H&E staining and quantification of histology score in GPIHBP1$^{-/-}$ mice (Scale bar = 100 μm, $n = 5$, $P = 0.0044$). **l** Flow cytometric quantification and statistical analysis of neutrophils infiltration in the pancreas of GPIHBP1$^{-/-}$ mice ($n = 5$, $P = 0.0014$). **m.** Representative immunofluorescence images and quantification of CitH3$^+$Mpo$^+$ cells in the pancreas (Scale bar = 50 μm, $n = 5$, $P = 0.0134$). **n** Serum CitH3 levels tested by ELISA ($n = 5$ biologically independent animals per group, $P = 0.015$). $P$ values were determined by two-tailed ordinary one-way ANOVA with the Tukey post hoc test or Student's *t*-test. Data was represented as mean ± SEM. *$p < 0.05$, **$p < 0.01$, ***$p < 0.001$. Source data are provided as a Source data file.

same dose of saline[36]. Mice were euthanized 11 h after the first injection.

### Antibiotic treatment (ABX) experiment

Gut microbiota was depleted by antibiotics treatment (ABX). Briefly, mice were treated with ABX (streptomycin (5 mg/ml) and clindamycin (0.1 mg/ml)) for seven days by drinking water[54]. The drinking water was replaced twice per week.

### Fecal microbiota transplantation (FMT)

Fresh fecal samples were collected from six HTGP patients, six HTG controls and six healthy volunteers (Donors had matched age and gender in each group). BMI- and serum triglycerides level-matched HTGP patients and high triglycerides controls were recruited (supplemental Table 2). Fecal slurry from each donor was colonized to three mice, and three recipient mice from each donor were placed in a cage. In detail, fecal microbiota suspensions were prepared by diluting and mixing 1 g of fecal samples obtained from donor in 10 ml of sterile PBS, shaken for 3 min, and then filtered through a 100 μm pore mesh. After 7 days of antibiotic treatment, the supernatant was delivered to the recipient mice via oral gavage (200 μL each recipient) within 10 min to prevent changes in bacterial composition (three times a week for three weeks). The control mice were orally gavaged with sterile PBS.

### Metabolomic analysis

Mice serum samples were collected and transferred to a fresh Eppendorf tube and centrifuged at $15,000 \times g$, 4 °C for 20 min. The supernatant was injected into the LC-MS/MS system analysis. UHPLC-MS/MS analyses were performed using a Vanquish UHPLC system (ThermoFisher, Germany) coupled with an Orbitrap Q ExactiveTM HF mass spectrometer (Thermo Fisher, Germany). Samples were injected onto a HypesilGoldcolumn ($100 \times 2.1$ mm, 1.9 μm) using a 17-min linear gradient at a flow rate of 0.2 mL/min. The raw data files generated by UHPLC-MS/MS were processed using the Compound Discoverer 3.1 (CD3.1, ThermoFisher). Statistical analyses were performed using the statistical software R (R version R-3.4.3), Python (Python 2.7.6 version) and CentOS (CentOS release 6.6). Metabolites were annotated using the KEGG database, HMDB database and LIPIDMaps database.

### Correlation analysis of gut microbial species and metabolites

Spearman's correlation of differentially enriched species and metabolites was calculated. Heatmaps were hierarchically clustered to represent the species-metabolite-associated patterns based on the correlation distance. All analyses and visualizations were implemented in python (v2.7.9) with the numpy (v1.9.2), scipy (v0.15.1), and matplotlib (v1.4.3) packages.

### Targeted amino acid profiling

For fecal samples, 50 mg of lyophilized feces was homogenized with 600 μl of ultrapure water. Amino acids (isoleucine, leucine, valine, phenylalanine, tyrosine, taurine and serine) were quantified by GC–MS. Aliquots of 50 μl serum were precipitated with 200 μl of methanol followed by vortex mixing for 5 min before centrifuged at $13,000 \times g$ for 10 min at 4 °C. Supernatant (200 μl) was transferred to a clean Eppendorf tube and dried under a gentle stream of nitrogen gas. The residue was derivatized by addition of 40 μl of methoxyamine hydrochloride (15 mg/ml in pyridine, 1 h at 60 °C) followed by trimethylsilyl derivatization using MSTFA [N-methyl-N-(trimethylsilyl) trifluoroacetamide 60 μl, 1 h at 70 °C]. After cooling down to room temperature, the mixture was centrifuged at $12,000 \times g$ for 10 min before GC–MS analysis.

### Flow cytometry for pancreatic immune cells

Pancreatic immune cells were isolated using collagenase IV digestion method for flow cytometry[55]. Splenocytes were separated through a 70 μm cell strainer. After washing and lysis of erythrocytes, $1 \times 10^6$ cells were preincubated with Fc block (Invitrogen, Cat.14-0161-81). For surface markers staining, cells were stained with FVD (Thermo Fisher Scientific, Cat.L34955), CD45 (BioLegend, Cat.103132), CD11b (BioLegend, Cat.101217), F4/80 (BioLegend, Cat.123141), Ly6g (BioLegend, Cat.127607), Ly6c (BioLegend, Cat.128015) antibodies. Intestinal lamina propria cells were isolated using 2 mg/mL collagenase D (Roche, Basel, Switzerland), 1 mg/mL dispase (Invitrogen, Carlsbad, CA), and 15 mg/mL DNase I for 60 min in a 37 °C shaking water bath for flow cytometry[56]. Cells were stained with FVD (Thermo Fisher Scientific, Cat.L34955), CD45 (BioLegend, Cat.103132), CD3 (BioLegend, Cat.100234), CD4 (BioLegend, Cat.100555), FOXP3 (Thermo Fisher Scientific, Cat.12-5773-82), ROR gamma (t) (Thermo Fisher Scientific, Cat.17-6988-82), anti-mouse IL-17A (BioLegend, Cat.506928) antibodies. After fixation and permeabilization cells were stained for the transcription factors (BioLegend, Cat.424401). Cells were incubated with Fc block and stained in different combinations of fluorochrome-conjugated antibodies. For intracellular staining, cells were fixed and permeabilized with the Foxp3/Transcription Factor Staining Buffer Set (eBioscience). Sample acquisition was carried out on BD FACS Celesta (BD Biosciences, San Jose, CA) and analysis was performed with FlowJo 10.8.

### Microbial strain

*Bacteroides uniformis* ATCC 8492 was purchased from American type culture collection (ATCC, USA). V4 of 16 S ribosomal RNA sequencing was performed to confirm bacterial strain at the species level. *Bacteroides uniformis* cultured in 1490 medium under anaerobic conditions using an anaerobic chamber. *Bacteroides uniformis* ($1 \times 10^8$ CFU /200 μL per mouse) was administrated by oral gavage every other day for 3 weeks[57]. Equivalent sterile PBS was used as vehicle control. The colonization of *Bacteroides uniformis* was confirmed by qRT-PCR. *Bacteroides uniformis*-treated recipient mice were housed with three or four animals per cage. The *B.uniformis*-treated and control mice were separately housed.

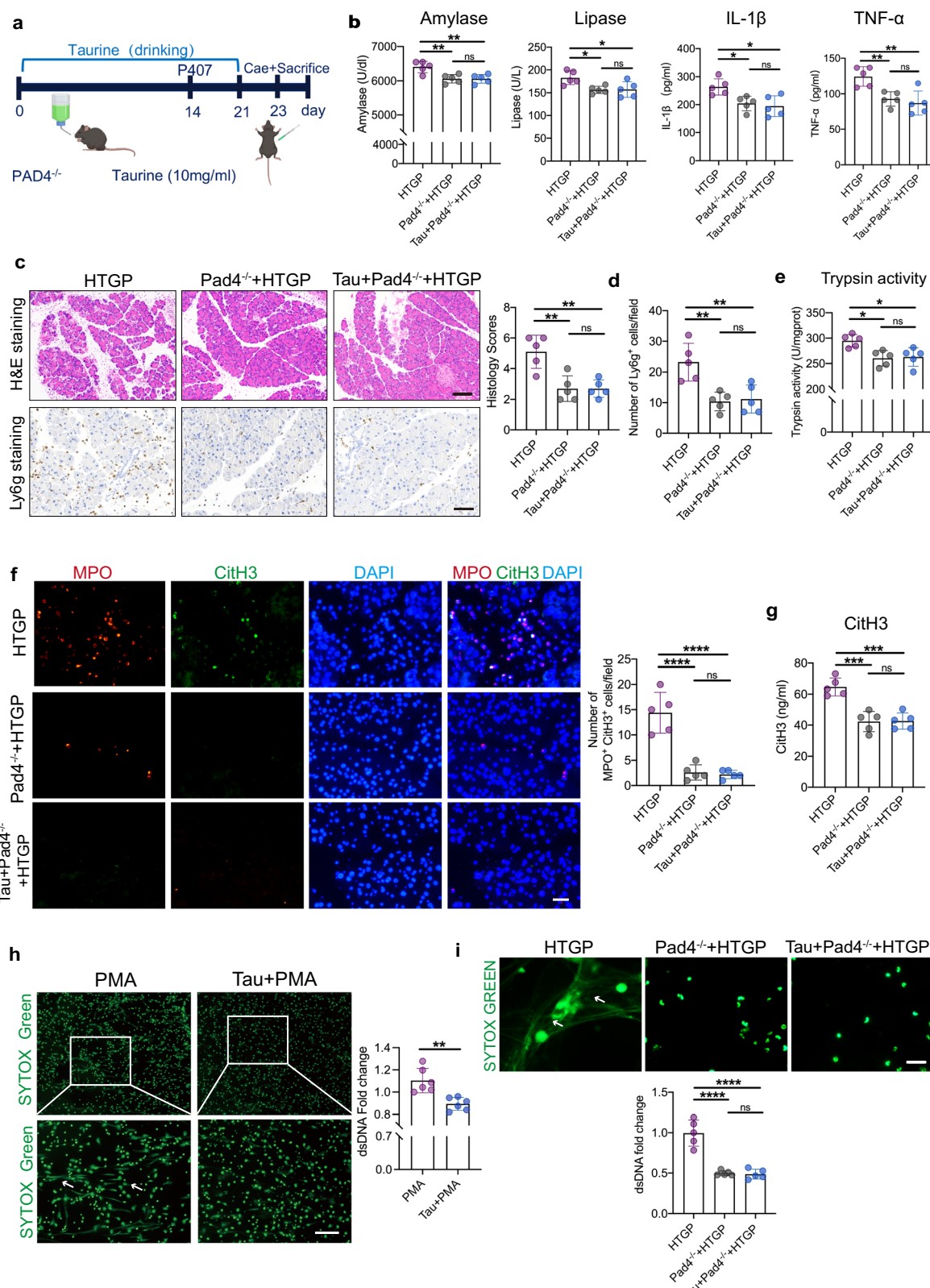

## Taurine supplementation

Taurine (10 mg/ml, Sigma-Aldrich Co., USA) was given prior to Caerulein injection for 3 weeks through drinking water[58]. Sterile water was given as vehicle control.

## ELISA

Mice plasma TNF-α, IL-1β and IL-17A levels were measured using ELISA kits (CUSABIO, Wuhan, China). Serum taurine, NETs and CitH3 levels were determined by specific enzyme-linked immunosorbent assay kit (Meimian, Jiangsu, China) and detected by Thermo Scientific Varioskan Flash.

## Immunofluorescence staining

Tissue was embedded in paraffin and sectioned at 4-μm thickness. Antigen retrieval was performed using target retrieval solution, and non-specific binding was then blocked with 1% BSA for 25 min.

**Fig. 6 | Taurine alleviates HTGP through NETs inhibition. a** Scheme for taurine supplementation. WT mice and PAD4$^{-/-}$ mice were administrated with sterile water (vehicle) or taurine supplementation for 3 weeks (HTGP, $n = 5$; HTGP in Pad4$^{-/-}$ mice, $n = 5$; Tau-treated HTGP in Pad4$^{-/-}$ mice, $n = 5$). **b** Serum amylase, lipase, IL-1β and TNF-α levels ($n = 5$ biologically independent animals per group) (Amylase: HTGP vs Pad4$^{-/-}$ + HTGP, $P = 0.0059$; Lipase: HTGP vs Pad4$^{-/-}$ + HTGP, $P = 0.0273$; IL-1β: HTGP vs Pad4$^{-/-}$ + HTGP, $P = 0.0277$; TNF-α: HTGP vs Pad4$^{-/-}$ + HTGP, $P = 0.0094$). **c** Representative images of H&E staining and quantification of histology score (Scale bar = 100 μm, $n = 5$ biologically independent animals per group) (HTGP vs Pad4$^{-/-}$ + HTGP, $P = 0.0022$; HTGP vs Tau+Pad4$^{-/-}$ + HTGP, $P = 0.0022$). **d** Representative images and quantification of Ly6g positive cells by IHC (Scale bar = 50 μm, $n = 5$) (HTGP vs Pad4$^{-/-}$ + HTGP, $P = 0.003$; HTGP vs Tau+Pad4$^{-/-}$ + HTGP, $P = 0.0048$). **e** Intrapancreatic trypsin activation ($n = 5$ biologically independent animals per group) (HTGP vs Pad4$^{-/-}$ + HTGP, $P = 0.0109$; HTGP vs Tau

+Pad4$^{-/-}$ + HTGP, $P = 0.0175$). **f** Representative immunofluorescence images and quantification of Mpo$^+$ CitH3+ cells in the pancreas (Scale bar = 50 μm, $n = 5$) (HTGP vs Pad4$^{-/-}$ + HTGP, $P < 0.0001$; HTGP vs Tau+Pad4$^{-/-}$ + HTGP, $P < 0.0001$). **g** Serum CitH3 levels tested by ELISA ($n = 5$ biologically independent animals per group) (HTGP vs Pad4$^{-/-}$ + HTGP, $P = 0.0002$; HTGP vs Tau+Pad4$^{-/-}$ + HTGP, $P = 0.0002$). **h** SYTOX Green fluorescence staining of neutrophil extracellular dsDNA after 3 h stimulation of PMA in the presence or absence of taurine (Scale bar = 50 μm, $n = 6$ biologically independent samples per group, $P = 0.002$). **i** Representative immunofluorescence staining images of SYTOX to map NETs in the neutrophils (Scale bar = 50 μm, $n = 6$ biologically independent samples per group) (HTGP vs Pad4$^{-/-}$ + HTGP, $P < 0.0001$; HTGP vs Tau+Pad4$^{-/-}$ + HTGP, $P < 0.0001$). $P$ values were determined by two-tailed ordinary one-way ANOVA with the Tukey post hoc test or Student's $t$-test. Data was represented as mean ± SEM. *$p < 0.05$, **$p < 0.01$, ***$p < 0.001$. Source data are provided as a Source data file.

Subsequently, tissues were incubated with CitH3 (1:100, Abcam, Cat.ab5103), Mpo (1:500, R&D, Cat.AF3667), Ly6g (1:800, BioLegend, Cat.127602) and anti-CD45 (1:200, Cell Signaling Technology, Cat.13917) overnight at 4 °C. Tissues were incubated with Alexa-Fluor-conjugated secondary antibodies (Invitrogen) for 1 h and DAPI was used for counterstaining the nuclei. Images were captured by confocal microscope (Nikon MODEL ECLIPSE Ni-E).

### Scanning electron microscopy (SEM)
NETs formation in pancreatic tissue samples was examined by high resolution scanning electron microscopy. Pancreatic tissues were fixed with 2.5% glutaraldehyde and 50-μm sections were embedded in resin. Sections were stained by lead citrate and uranyl acetate and captured by electron microscope (ZEISS, Sigma 300).

### RNA sequencing analysis
In our study, pancreas tissues from FMT-HC and FMT-HTGP mice were harvested after inducing HTGP model. RNAs from three biological replicates of each group were isolated using RNeasy kit (Qiagen China (Shanghai) Co Ltd, China), and total amounts and integrity of RNA were assessed using the RNA Nano 6000 Assay Kit of the Bioanalyzer 2100 system (Agilent Technologies, CA, USA). The gene expression profiles of FMT-HC and FMT-HTGP groups were investigated using the Illumina NovaSeq 6000 according to the manufacturer's guide (Illumina, San Diego, CA, USA). The detailed description and plot visualization of RNA-seq quality control were uploaded to the Supplementary table 3.

### Tissue dissociation and single-cell RNAseq analysis
Pancreas samples from FMT-HC and FMT-HTGP mice were collected. Tissues were washed with Hanks Balanced Salt Solution (HBSS), and digested in 2 ml GEXSCOPE Tissue Dissociation Solution (Singleron Biotechnologies) following manufacturer's instructions[59].

### Single-cell RNA-sequencing library preparation
Briefly, the specimens were digested at 37 °C for 15 min and passed through 40-micron sterile strainer (Corning). Cells were centrifuged at $300 \times g$ for 5 min and cell pellets were resuspended in 1 ml PBS (HyClone). Cell suspensions were counted to determine cell viability, and the concentration was adjusted to $1 \times 10^5$ cells/ml. Single-cell suspension was then loaded to microfluidic chip (GEXSCOPE Single Cell RNA-seq Kit, Singleron Biotechnologies) and scRNA-seq libraries were constructed according to the manufacturer's instructions (Singleron Biotechnologies). Libraries were sequenced on Illumina Novaseq 6000 platform with 150 bp paired end reads.

### Primary analysis of raw read data
Raw reads were processed to generate gene expression profiles using CeleScope v1.5.2 (Singleron Biotechnologies) with default parameters. Briefly, Barcodes and UMIs were extracted from R1 reads and

corrected. Adapter sequences and poly A tails were trimmed from R2 reads and the trimmed R2 reads were aligned against the GRCm38 (mm10) transcriptome using STAR (v2.6.1b). Uniquely mapped reads were then assigned to exons with FeatureCounts (v2.0.1). Successfully Assigned Reads with the same cell barcode, UMI and gene were grouped together to generate the gene expression matrix for further analysis.

### Quality control, dimension-reduction and clustering (Seurat)
Seurat v 3.1.2 was used for quality control, dimensionality reduction and clustering. For each sample dataset, the expression matrix was filtered using the following criteria: (1) cells with gene count less than 200 or with top 2% gene count were excluded; (2) cells with top 2% UMI count were excluded; (3) cells with mitochondrial content > 30% were excluded; (4) genes expressed in less than 5 cells were excluded. After filtering, 28,638 cells were retained for the downstream analyses, with on average 2622 genes and 9692 UMIs per cell. Gene expression matrix was normalized and scaled using functions NormalizeData and Scale-Data. Top 2000 variable genes were selected by FindVariableFeatures for PCA analysis. Cell clusters were visualized using t-Distributed Stochastic Neighbor Embedding (t-SNE) or Uniform Manifold Approximation and Projection (UMAP) with Seurat functions RunTSNE and RunUMAP. The cells were filtered by UMI counts, gene counts, and median genes per cell, which were all shown in Supplementary Table 4.

### Differentially expressed genes (DEGs) analysis
To identify differentially expressed genes (DEGs), we used the Seurat FindMarkers function based on Wilcoxon rank sum test with default parameters, and selected the genes expressed in more than 10% of the cells in both of the compared groups of cells and with an average log (Fold Change) value greater than 0.25 as DEGs. Adjusted $P$ value was calculated by Bonferroni Correction and the value 0.05 was used as the criterion to evaluate the statistical significance.

### Pathway enrichment analysis
To investigate the potential functions of immune-relevant genes, Gene Ontology (GO) and Kyoto Encyclopedia of Genes and Genomes (KEGG) analysis were used with the "clusterProfiler" R package v 3.16.1. Pathways with $P\_$adj value less than 0.05 were considered as significantly enriched. Selected significant pathways were plotted as bar plots. Gene Set Enrichment Analysis (GSEA) was conducted using clusterProfiler package (version 3.5). The fold change of gene expression between FMT-HC group and FMT-HTGP group was calculated, and the gene list was generated according to the change of |log2FC|. For GSVA pathway enrichment analysis, the average gene expression of each cell type was used as input data.

### Cell-type recognition with Cell-ID
Cell-ID is multivariate approach that extracts gene signatures for each individual cell and perform cell identity recognition using

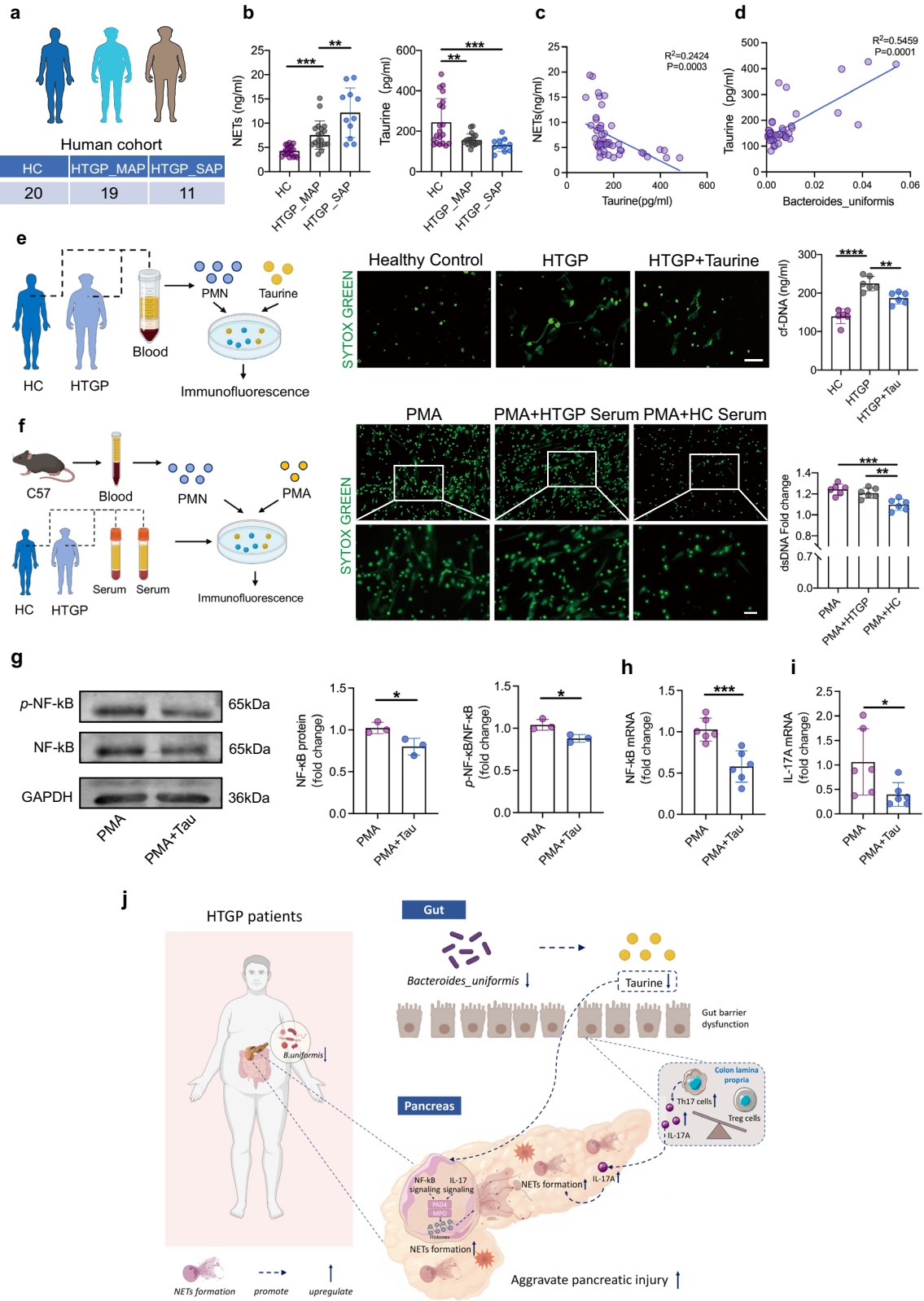

hypergeometric tests (HGT). Dimensionality reduction was performed on normalized gene expression matrix through multiple correspondence analysis, where both cells and genes were projected in the same low dimensional space. Then, a gene ranking was calculated for each cell to obtain most featured gene sets of that cell. HGT were performed on these gene sets against brain reference from SynEcoSys database, which contains all cell-type's featured genes. Identity of each cell was determined as the cell-type has the minimal HGT $P$ value. For cluster annotation, Frequency of each cell-type was calculated in each cluster, and cell-type with highest frequency was chosen as cluster's identity[60].

The cell type identification of each cluster was determined according to the expression of canonical markers from the reference database SynEcoSysTM. SynEcoSysTM contains collections of canonical cell type markers for single-cell seq data, from CellMakerDB, PanglaoDB and recently published literatures.

**Fig. 7 | Taurine blocks NETs and restrains inflammatory response in HTGP patients. a** Information of HC ($n = 20$) and HTGP patients, including HTGP_MAP ($n = 19$) and HTGP_SAP ($n = 11$), according to the 2012 revised Atlanta criteria. **b** Correlation analyses between serum NETs levels and the severity of HTGP, and serum taurine levels and disease severity (HC, $n = 20$; HTGP_MAP, $n = 19$; HTGP_SAP, $n = 11$) (NETs: HC vs HTGP_MAP, $P = 0.0042$, HTGP_MAP vs HTGP_SAP, $P = 0.0006$; Taurine: HC vs HTGP_MAP, $P = 0.003$, HC vs HTGP_SAP, $P = 0.001$). **c** Correlation analysis between serum taurine and NETs levels. **d** Correlation analysis between taurine level and *Bacteroides uniformis* abundance. **e** Representative immuno-fluorescence staining images of SYTOX to map NETs in the neutrophils isolated from HTGP patients and healthy volunteers with PMA or taurine stimulation (Scale bars = 50 μm, $n = 6$) (HC vs HTGP, $P < 0.0001$; HTGP vs HTGP + Tau, $P = 0.0061$). Quantification of NETs release by measuring supernatant DNA level. **f** Representative images of SYTOX to map NETs in mice neutrophils stimulated by PMA, in the presence of healthy donors' serum or HTGP patients' serum (Scale bars = 25 μm, $n = 6$) (PMA vs PMA + HC, $P = 0.0006$; PMA + HTGP vs PMA + HC, $P = 0.0049$). **g** Representative western blot images and quantitative comparison of total and phosphorylation of NF-κB protein within neutrophils stimulated by PMA or PMA + Tau ($n = 3$ biologically independent samples per group) (NF-κB protein, $P = 0.0338$; Phosphorylation of NF-κB protein, $P = 0.0237$). **h** The relative *Nfkb* mRNA expression in the neutrophils stimulated by PMA or PMA + Tau ($n = 6$ biologically independent samples per group, $P = 0.0009$). **i** The relative *Il17a* mRNA expression in the neutrophils stimulate by PMA or PMA + Tau ($n = 6$ biologically independent samples per group, $P = 0.047$). **j** Gut microbiota aggravates neutrophil extracellular traps-induced pancreatic injury in HTGP. $P$ values were determined by two-tailed ordinary one-way ANOVA with the Tukey post hoc test or Student's $t$-test. Data was represented as mean ± SEM. $^*p < 0.05$, $^{**}p < 0.01$, $^{***}p < 0.001$. Source data are provided as a Source data file.

## Subtyping of major cell types

To obtain a high-resolution map of Mononuclear phagocytes(MPS), Fibroblasts, Acinar cells, Neutrophils, PancreaticDuctal, Erythrocytes, PlasmaCells, PanSCs, NeuroendocrineCells, MesothelialCells, Endothelial cells(ECs), T cells and B cells, cells from the specific cluster were extracted and re-clustered for detailed analysis following the same procedures described above and by setting the clustering resolution as 0.8.

## Filtering cell doublets

Cell doublets were estimated based on the expression pattern of canonical cell markers. Any clusters enriched with multiple cell type-specific markers were excluded for downstream analysis.

## UCell gene set scoring

Gene set scoring was performed using the R package UCell v 1.1.0[61]. UCell scores are calculated by ranking the relative query genes levels of individual cells using Mann–Whitney $U$ test. Since UCell is a rank-based scoring method, it is suitable to be used in large datasets containing multiple samples and batches.

## NETosis assay

Freshly isolated peripheral neutrophils were seeded in 24-well plate ($1 \times 10^5$ cells per well) and treated with 100 nM 12 phorbol 13-myristate acetate (PMA) (Sigma-Aldrich, St. Louis, MO, USA) for 3 h. For serum-based NETosis assays, 10% serum from healthy volunteers or HTGP patients was added to neutrophils with PMA. Finally, 200 nM SYTOX Green (Invitrogen) was carefully added to the plate without disturbing the NETs. Images were captured by fluorescence microscope after 15 min. Supernatants containing the NETs fragments were collected and centrifuged (10 min, $1000 \times g$) to remove cell debris. NETs were quantified using a Quant-iT Picogreen dsDNA Assay Kit (Invitrogen, P11496) according to the manufacturer's instructions[62].

For human studies, peripheral neutrophils were collected from healthy volunteers and HTGP patients. Freshly human peripheral neutrophils were seeded in each well of a 24-well plate ($5 \times 10^4$ cells per well) and taurine (40 mM) was added. After 3 h, 200 nM SYTOX Green was carefully added to the plate to detect NETs by fluorescence microscope.

## Histopathological and immunohistochemical analysis

Tissues were fixed and embedded, 4-μm sections were stained with H&E. Pancreatic damage was assessed by two pathologists in a blinded manner. Briefly, evaluation of pancreatic pathology included four categories: edema, inflammatory cell infiltration, necrosis and vacuolization[63]. The Mpo (1:1000, Abcam, Cat.ab208670) and Ly6g positive cells (1:250, Cell Signaling Technology, Cat.87048 S) were tested by immunohistochemistry and quantified by Image J.

## Real-time quantitative PCR

The tissues were homogenized and total RNA was isolated by Trizol Reagent (Invitrogen, USA). Concentration was measured using the NanoDrop (Thermo Fisher Scientific, USA). Real-time qPCR was performed using the ABI 7500 real-time PCR system (Applied Biosystems). Specific primers for quantitative PCR used in this study are shown in Supplementary Table 5.

## Western blot analysis

Whole-cell lysates with approximately 40 μg of proteins were resolved on 10% SDS-PAGE and were subjected to western blot assay using the anti-NF-κB p65 (Cell Signaling Technology, Cat.8242, 1:1000) and anti-phospho-NF-κB p65 (Cell Signaling Technology, Cat.3303, 1:1000). After appropriate secondary antibody incubation, the bands were visualized with the Molecular Imager System (BIO-RAD, Hercules, USA) using an enhanced chemiluminescence method (Thermo Fisher Scientific, Massachusetts, USA)[64].

## Myeloperoxidase and trypsin activity

Lung Mpo activity was determined using Mpo Detection Kit (Nanjing Jiancheng Bioengineering Institute, China). Pancreas trypsin activity was determined using Trypsin assay kit (Nanjing Jiancheng Bioengineering Institute, China) and detected by Thermo Scientific Varioskan Flash.

## Detection of bacteria colonization in mice

The fecal bacteria solution was stained with 10 μg/ml CM-DiI fluorescent dye (Thermo Fisher Scientific, Waltham, MA, USA) for 30 min at 37 °C. Mice were gavaged with 200 μl stained fecal bacteria solution. Fresh feces of recipient mice were collected on seven days after intragastric administration. The feces were examined by confocal microscope (Nikon MODEL ECLIPSE Ni-E).

## Preparation of human peripheral neutrophils

Neutrophils were collected from peripheral anticoagulated blood samples after density gradient centrifugation. A sterile processing environment was maintained with a clean bench, and 5 ml whole blood sample was separated in aliquots into 15 mL test tubes with 5 ml Polymorphprep (Axis-Shield, Oslo, Norway), followed by centrifugation for approximately 30 min at 500 g. The granulocyte layer was carefully removed and resuspended in RPMI 1640 media plus 25 mM HEPES (Gibco) and centrifuged at $500 \times g$ for 5 min to remove any remaining Polymorphprep. Cells were resuspended in media and contaminating erythrocytes were removed by hypotonic lysis. Platelet contamination was removed by further centrifugation of the cells at $150 \times g$ for 3 min. Neutrophils were resuspended in RPMI 1640 media at a final concentration of $5 \times 10^6$/ml.

## Neutrophil depletion

Neutrophils were depleted by intraperitoneal injection of two doses (200 μg/mouse, 24 h before Caerulein injection; 100 μg/mouse, 4 h

before Caerulein injection) of anti-mouse Ly6G (1A8, BioXCell)[65]. The rat IgG2a isotype was used as control (2A3, BioXCell).

## Statistical analysis

Samples sizes were determined using comparison to prior experience, but no statistical method was used to determine sample size. All error bars represent the standard error of the mean (SEM). Statistical significance was assigned at $P$ values of <0.05 and detected by GraphPad Prism 9.0 (GraphPad Software, San Diego, USA). The Shapiro-Wilk test was used to determine the sample distribution type. A two-tailed Student's $t$-test was used to evaluate statistical significance between two groups for normal distribution. For the nonparametric tests, the two-tailed Mann-Whitney test was used to evaluate statistical significance between two groups. For more than two groups, the significance was calculated by the Kruskal-Wallis test or ordinary one-way ANOVA depending on the sample distribution type; a post hoc Tukey test was used to conduct multiple comparisons. Bray-Curtis dissimilarity matrices were calculated and used for ordination by PCoA. These matrices were also used to assess differences in beta diversity by Anosim. PCoA was carried out for all dimension reduction analyses using the vegan package in R software. Correlations between enriched species and circulating metabolites were tested with Spearman's correlation. For clinical data analysis, parametric test (one-way ANOVA) was used to compare the data between groups. Categorized variables were compared using chi-square test or the Fisher exact test, as appropriate. Statistical analyses were conducted using SPSS v.26.0 (IBM Corporation). The statistical test used, and $P$ values are indicated in each figure legend. $P$ values of <0.05 were considered statistically significant. *$P$ < 0.05, **$P$ < 0.01, ***$P$ < 0.001 and ****$P$ < 0.0001. No data were excluded from the analyses. The mice were randomly divided into experimental groups, and investigators were not blinded to allocation during experiments and outcome assessment.

## Reporting summary

Further information on research design is available in the Nature Portfolio Reporting Summary linked to this article.

## Data availability

All 16 S rRNA gene sequences were provided and available at National Center for Biotechnology Information Sequence Read Archive (SRP) database with accession code PRJNA950569. The single-cell data generated in this study have been deposited in the Genome Sequence Archive in National Genomics Data Center under accession code BioProject: PRJCA016154/CRA012178 [https://ngdc.cncb.ac.cn/bioproject/browse/PRJCA016154]. The metabolic data and RNA sequencing data generated in this study have been deposited in the Genome Sequence Archive in National Genomics Data Center under accession code BioProject: PRJCA016154 (OMIX003700 and OMIX003704). Source data are provided with this paper.

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

## Acknowledgements

We show our full respect and gratitude to all the participants in the study. This study was supported by The National Natural Science Foundation of China (Nos. 82270665 (B.S.); Nos. 82070658 (B.S.); Nos. 82270666 (L.L.); Nos. 81871974 (B.S.); Nos. 81702384 (Y.W.)). The Science Fund for Excellent Young Scholars of First Affiliated Hospital of Harbin Medical University (Nos. HYD2020YQ0009 (L.L)). The Open Fund of Key Laboratory of Hepatoaplenic Surgery, Ministry of Education, Harbin, China (Nos. GPKF202201 (G.L.)). The Natural Science Foundation of Heilongjiang Province (Nos. TD2021H001 (B.S.)). The Youth Innovation Talent Training Program of the General Undergraduate Colleges and Universities in Heilongjiang province (Nos. UNPYSCT-2020157 (L.L)). Schematic diagram was conducted using BioRender.

## Author contributions

G.Q.L., L.L. and B.S. conceptualised and designed the study. T.Q.L., Y.X., P.X., H.Z.C., G.W., Y.W.W., R.K., H.C., D.B.X., H.T.T., Z.B.L., X.W.B. and

J.S.H. were involved in collection, preparation, interpretation, validation and critical review of the data. Z.J.Z., C.Z. and C.D.C. performed the cell culture and western blot experiments. G.Q.L., L.L., T.Z. and Y.H.S. performed animal experiments. G.Q.L. and Y.H.S. performed histological evaluation. G.Q.L. and L.W.L. performed formal analysis including bioinformatics and statistical analyses. G.Q.L. created the manuscript figures and supplementary materials. G.Q.L. and L.L. drafted the manuscript. F.M. edited and reviewed the manuscript text. B.S. and L.L. acquired funding for the study.

## Competing interests

The authors declare no competing interests.
