## [Peer Review File · Nature Communications]

REVIEWER COMMENTS

Reviewer #1 -Microbiome / FMT / metabolomics- (Remarks to the Author):

The authors describe a study primarily comparing the intestinal microbiota among hypertriglyceridemic pancreatitis (HTGP), HTG, and healthy participants. Serum metabolomics and immune profiling were done. Results were further tested using murine models with reciprocal FMT. *Bacteroides uniformis* and taurine were found to have beneficial effects. I commend the authors on generating a comprehensive dataset to inform microbiota-metabolite-immune-host interactions. Considerable revision is needed by a native English speaker. Throughout the results, the authors tend to overstate conclusions based on their own data and they acknowledge trends in the literature (e.g., F/B) ratio that have been largely discredited. There seems to be an underlying assumption that the microbiota are the primary causative driver of disease, although there are no data to support that directionality. This in no way diminishes the fine work the authors have done exploring mechanism associated with disease. Along the same lines, the PICRUSt piece could be removed. The authors have actual metabolomic data to support their conclusions; there is no need for inferred function that is autocorrelated with taxonomy. More details on mouse housing (individually, cohoused) are needed to better assess scientific rigor.

Specific comments:

Supplementary Table 1 could not be found.

pg 6, line 2, please provide the statistical test, effect size, and P-value used to evaluate significance.

pg 7, lines 6 and 8: Please define SIRS and CRP at first use.

pg 7, lines 10-12: This is a considerable overstatement of conclusions drawn from correlative data from a small cohort.

pg 8, line 8: Please indicate which statistic was used and the effect size.

pg 8, line 11: Please specify "potentially" pathogenic as pathogenicity was not determined.

pg 8, line 19: Pathogen colonization was not tested in this study. Please clarify.

pg 9, lines 7-10: What data do the authors' have to support these statements?

pg 15, line 1: What is PMA?

pg 22, line 14: What number(s) were samples normalized to?

pg 25, line 6: What are the demographics of the donors? Were they matched?

pg 25, line 9: How much material was used for each gavage (vol and number of cells)?

Fig 1, panel D: Please correct axis label to ASV for consistency.

STORMS checklist:

1.2: The type of sequencing is not described in the abstract.

1.3: Sample types should be described in the abstract.

3.6: Description of the sample size is not justified.

4.8: Negative controls not described for PCR reactions.

4.10: Mouse housing needs to be described to understand rigor of replication.

6.0: Discussion should acknowledge that only male mice were used.

7.1: 16S rRNA amplicon processing should also be noted in checklist

8.1: Raw data must be deposited in a public repository and an accession number(s) provided.

<https://www.nature.com/sdata/policies/repositories>

8.2; The current statement in 8.1 applies to 8.2

13.0/1: Limitations and bias need to be addressed.

Reviewer #2 -Pancreatitis pathology / neutrophils / animal models- (Remarks to the Author):

Gut microbiota aggravates neutrophil extracellular traps-induced pancreatic injury in hypertriglyceridemic pancreatitis

The authors show on the basis of human volunteers and patients that gut dysbiosis is present in hypertriglyceridemic pancreatitis patients (HTGP). Especially *Bacteroides uniformis* is reduced. The authors also find a correlation between the presence of *Bacteroides uniformis* and the metabolite taurine and can show in animal studies that taurine is able to suppress net formation of neutrophils and thus reduce the severity of the disease.

Recent studies clearly show a dysbiosis in patients with acute and chronic pancreatitis, especially facultative pathogens increase and displace supposedly protective strains such as *Bacteroides uniformis*. However, why patients with HTGP were used and not acute pancreatitis patients in general is not clear to me. The mechanism described is probably not specific for HTGP patients. It is widely known that BMI is associated with serum triglycerides and changes of the gut microbiome.

A more general approach to the role of the microbiome would be more useful and a general interest. A detailed breakdown of the mechanism would also significantly strengthen the manuscript. Besides neutrophils, monocytes play a much more important role in pancreatic damage.

Major points:

The authors compare healthy volunteers (HC), high triglycerides (HTG) and hypertriglyceridemic pancreatitis patients (HTGP) with regard to their faecal microbiome. However, in order to draw conclusions about the effect of the microbiome on pancreatitis, especially the effect of high triglycerides, the comparison to the microbiome of patients with acute pancreatitis without hypertriglyceridemia is missing.

Figure 2 shows nicely the animal experiments and the faecal transfer from patients. Here I am a bit confused. Authors proclaim 3 mice by each donor (6/group). So, 18 animals per group, but the graphs show only 6-8 mice in one group, why were more than half of the mice excluded? Here also controls missing without pancreatitis induction. Does the simple Transfer of microbiome affect the organ?

Does FMT affect intestinal barrier and immune response? Th17/Treg axis is essential for the immunological arm of the intestinal barrier, and it is known that IL17A is released from Th17 cells and ILC3s in the lamina propria and is able to induce NET formation (PMID: 26964500).

What exact test was used for Figure 2C. The standard deviation is quite high, what correction method was used for multiple testing. This point is not very clear in Material and Methods.

Figure 2C shows LPS which is not mentioned in the text or the figure legend. Did the authors proclaim that the animals develop an infected necrosis? If yes, then this point needs to be clarified by additional experiments to show which bacteria infiltrate the pancreas.

In general, the FMT experiments with faecal samples from patients with acute pancreatitis without hypertriglyceridemia are missing and would strongly support the story.

Authors found a more pronounced neutrophil and monocyte/macrophage infiltration in the pancreas of FMT-HTGP mice, both are increasing in the same manner. Why do authors focus on neutrophils instead of macrophages. It is known that macrophages are key cells for inflammatory response in pancreatitis (PMID: 32739869). On the other hand, it is known that neutrophils can induce intrapancreatic protease activation via release of ROS (PMID: 11910350), could authors measure increased trypsinogen activation in the pancreas of FMT-HTGP mice?

Figure 5 J-M is missing in the result section. Figure 5K shows flow cytometry of neutrophils isolated from the pancreas. It seems that Taurine treated mice have a much higher number of leukocytes which leads to a lower percentage of neutrophils. What about macrophages and monocytes? Additional plot with the cell count would verify the results.

Can the authors show that significant amounts of taurine reach the pancreas?

Disease severity in nearly all animal experiments is mainly shown by histological scoring. Could the authors also provide data such as amylase and lipase activity in serum, intrapancreatic protease activation as well as lung MPO as marker for systemic inflammation. The disease severity is the key

point in argumentation and need to be verified by different methods not only histology score. A clear evaluation of systemic immune response would help to understand the results better and would give an idea how NET formation influence disease severity.

Taurin is mentioned to reduce NET formation. A mechanism how Taurin acts on neutrophils should be shown. In Figure 7F the authors proclaim that Taurin from serum inhibits PMA induced NET formation, how does this work? A clear mechanism needs to be shown.

Figure 7G: The results do not support the figure presented here. It is conveyed that HTGP is induced by neutrophil activation and NET formation, this is simply false. Neutrophils influence the severity, not the onset of disease. It has been shown that the IL17-NET axis can lead to ductal obstruction, but for this to happen the cells must first migrate into the organ and be activated, i.e. the pancreatitis must have already begun. This figure need to be modified.

Minor points:

The first sentence of the introduction: „and the mortality rate ranges from 20% to 40%“. Such a dramatic mortality rate is only present in severe form of pancreatitis not in general.

Figure 1C. The authors should show the percentage of difference in the NMDS plot.

Can Authors show that the depletion of Gut microbiota in mice worked?

Co-housing and horizontal microbiome transfer plays an important role in mice experiments, could the authors explain their strategy.

In Material and Methods, caerulein was administered hourly for 10 h and the animals were sacrificed 11 h after the first application. In the overview chart (Figure 2A, 4B), however, 1 day after caerulein treatment. This need to be clarified, 12h or 24h after onset of disease makes a difference.

Figure 4G, the axis indicate a greater difference than is realistic, the y-axis should start at 0.

A detailed explanation of the statistical tests in the material and methods section is necessary.

Response to Reviewers

Dear Referees:

Thank you for all referees' comments concerning our manuscript entitled "Gut microbiota aggravates neutrophil extracellular traps-induced pancreatic injury in hypertriglyceridemic pancreatitis (NCOMMS-23-01234)". These comments were highly insightful and enabled us to improve the quality of our manuscript. We have studied all comments carefully and made revisions in accordance with the comments. All changes in the revised manuscript were highlighted with blue color. We hope that the revised edition will be sufficient to make our manuscript suitable for publication.

The following pages are point-by-point responses to each comment.

Reviewer #1

Q1.The authors describe a study primarily comparing the intestinal microbiota among hypertriglyceridemic pancreatitis (HTGP), HTG, and healthy participants. Serum metabolomics and immune profiling were done. Results were further tested using murine models with reciprocal FMT. *Bacteroides uniformis* and taurine were found to have beneficial effects. I commend the authors on generating a comprehensive dataset to inform microbiota-metabolite-immune-host interactions.

We really appreciate your interest for our study. Based on the comments below, we tried our best to improve our manuscript, and hope the revision will clarify the concerns for this study.

Q2. Considerable revision is needed by a native English speaker.

We firstly finished all the revision, and the manuscript was revised by a native
speaker which we hope may make it much readable and understandable.

**Q3.** Throughout the results, the authors tend to overstate conclusions based on their
own data and they acknowledge trends in the literature (e.g., F/B) ratio that have been
largely discredited.

For F/B ratio analysis, the standard deviation is high, suggesting a huge variation
of F/B ratio within groups. We understood that the ratio of F/B did not make any
conclusion and decided to remove it from the manuscript. The manuscript was revised
as "A At the phylum level, the decrease of *Firmicutes* and expansion of *Proteobacteria*
were identified in HTGP patients, compared with healthy volunteers ($P=0.003$)
(Supplementary Fig.1a,b and Fig.R1)." (Page 5 lines 15 to 18)

**Fig.R1 Compositional analysis of the gut microbiome of HTGP patients.**(a) Relative abundance of
gut microbiota in phylum level. Top 20 abundant and prevalent phyla were compared. (b) Relative
abundance of gut microbiota in *Firmicutes*.

**Q4.** There seems to be an underlying assumption that the microbiota are the primary
causative driver of disease, although there are no data to support that directionality.
This in no way diminishes the fine work the authors have done exploring mechanism

associated with disease. Along the same lines, the PICRUSt piece could be removed.
The authors have actual metabolomic data to support their conclusions; there is no
need for inferred function that is autocorrelated with taxonomy.

In current study, we have compared the differential expressed gut microbiota
between healthy population and hypertriglyceridemic pancreatitis (HTGP) patients.
The whole story was built beyond the alternation of gut microbiota in HTGP patients.
It is not correct to mention that particular bacteria promote the initiation of
pancreatitis or only participate in the development of disease. Since the deficiency of
spontaneous mouse model for studying the initiation of HTGP, our main goal was to
investigate the molecular mechanism of *B.uniformis* promoting taurine synthesis that
inhibits NETs formation in the pathogenesis of HTGP. In the revised manuscript, we
have removed all the PICRUSt analysis results following your suggestion.

**Q5.** More details on mouse housing (individually, cohoused) are needed to better
assess scientific rigor.

In our study, all animals were housed in specific pathogen-free environment with
controlled conditions, including a constant temperature ($22 \pm 2^\circ\text{C}$), 12-h light/12-h
dark cycle and $50 \pm 5\%$ humidity. The animals had free access to water and standard
chow diet (Xie Tong Biotechnology, SFS9112). The sterile drinking water was replaced
twice a week. The above animals were commercially available and acclimated for at
one week before experiment started. (Page 21 line 24, Page 22 lines 1 to 5)

For the FMT experiment, fresh fecal samples were collected from six HTGP
patients, six HTG controls and six healthy volunteers (The donors were age and gender
matched between groups). Fecal slurry from each donor was colonized to three mice,

and three recipient mice from each donor were housed in the same cage. We
deliberately did not mix these groups to avoid a possible horizontal microbiome
transfer. For preparing fecal microbiota suspensions, 1 g of fecal samples obtained
from donor were diluted in 10 ml of sterile PBS, vortexed for 3 min, and then filtered
through a 100 µm pore mesh. After 7 days of antibiotic treatment, the solution was
orally gavaged to the recipient mice (200 µL each recipient three times a week for
three weeks). (Page 22 lines 20 to 24, Page 23 lines 1 to 7)

For the *B.uniformis* administration experiment, all the recipient mice were
exposure to antibiotic treatment for one week and then were randomized divided into
*B.uniformis*-treated and control groups. Mice in each group were housed separately.
(Page 25 lines 19 to 21)

**Specific comments:**

**Q1.** Supplementary Table 1 could not be found.

Due to be uploaded as a Supplementary table, the Supplementary Table 1 was
not assessable as a pdf file. We have added Supplementary Table 1 as part of
manuscript, also listed it below for easily reviewing. We apologized for the issue for
the review process. (Page 5 line5)

Supplementary Table 1. Demographic and clinical characteristics of three groups.				
Characteristics	HC(n=16)	HTG(n=8)	HTGP(n=25)	P value
Male, n (%) [*]	9(56.25)	4(50.00)	14(56.25)	1.000
Age(years), mean (SD) [#]	40.94(±12.8)	41.38(±7.9)	39.48(±10.2)	0.871
Triglyceride(mmol/L),mean (SD) ^{##}	1.64(±0.88)	9.14(±2.41)	8.87(±2.87)	<0.001 [†]
BMI (kg/m ²), mean (SD) [#]	23.87(±2.47)	28.05(±1.34)	27.96 (±2.97)	<0.001 [†]
Drinking, n (%) [*]	5(31.25)	3(37.50)	9(36.00)	1.000
Smoking, n (%) [*]	5(31.25)	2(25.00)	8(32.00)	1.000
Hypertension, n (%) [*]	5(31.25)	2(25.00)	7(28.00)	0.902
Diabetes, n (%) [*]	6(37.50)	3(37.50)	11(44.00)	0.926

* Fisher's exact test; # One-Way ANOVA; ## Nonparametric rank sum test

† Indicated HC group was significantly different from the other two groups.

**Q2.** pg 6, line 2, please provide the statistical test, effect size, and P-value used to
evaluate significance.

In our study, non-metric multidimensional scaling (NMDS) was used to visualize
the differences of beta diversity between groups in complex multi-dimensional data^{1,2}.
NMDS was visualized by R package. NMDS usually judges the rationality of the sorting
model according to the Stress function value. If the Stress is lower than 0.2, it indicates
that NMDS can accurately reflect the degree of difference between samples.

In our study, NMDS analysis (Stress=0.139) showed significant difference on the
diversity of gut microbiota between three groups (Fig.1c, Fig.R2a). The analysis of
similarity (Anosim) indicated clear separation of the gut microbiota between HTGP
patients and healthy volunteers (Anosim; R=0.4928, P=0.001). Moreover, the
difference between HTG controls and HTGP patients was evaluated by analysis of

similarity (Anosim; $P < 0.05$) (Fig.R2b). These results indicate a less heterogeneous
community structure in HTGP patients compared with controls. (Page 5 lines 9 to 14)

**Fig.R2 The separation of the microbial composition.** (a) Nonmetric multidimensional scaling
(NMDS) plot with cluster indicating differential microbial distributions between three groups. (b)
The analysis of similarity (Anosim) was applied to evaluate the separation of gut microbiota
between HC, HTG and HTGP patients.

**Q3.** pg 7, lines 6 and 8: Please define SIRS and CRP at first use.

The definition of Systemic inflammatory response syndrome (SIRS) and C-reactive
protein (CRP) have been added at the first-time use. (Page 6 lines 6 to 8)

**Q4.** pg 7, lines 10-12: This is a considerable overstatement of conclusions drawn from
correlative data from a small cohort.

Recent studies revealed the features of gut microbiota have the potential to
become biomarkers for pancreatic diseases³. HTGP is characterized by younger age,
higher risks of infected necrosis, organ failure and death^{4,5}. Our current study aims to
explore the association between altered gut microbiota and the prognosis of HTGP

patients, which may elucidate the potential mechanisms of gut microbiota influencing
HTGP progression and unveil microbial biomarkers for further validation and utility. As
suggested, we revised the statement in the results as "These suggest that specific
bacterial species are associated with disease severity which may affect patients'
prognosis, indicating a potential pathophysiological link between altered gut
microbiota and HTGP development." (Page 6 lines 10 to 13)

**Q5.** pg 8, line 8: Please indicate which statistic was used and the effect size.

The Bray-Curtis distances were analyzed, and principal coordinate analysis (PCoA)
was used to visualize different beta diversity between groups. PCoA showed clear
separation of gut microbial characters between FMT-HC and FMT-HTGP mice (Fig.2e,
Fig.R3a). The analysis of similarity (Anosim) was applied to evaluated significant
separation of the gut microbiota of the FMT-HC and FMT-HTGP group (Anosim;
$R=0.4244$, $P=0.002$). Moreover, the gut microbiota community structures of the FMT-
HTG mice and FMT-HTGP mice were differed (Anosim; $R=0.404$, $P=0.001$) (Fig.R3b).
(Page 8 lines 4 to 6)

**Fig.R3 The gut microbial characters between FMT-mice.** (a) Principal coordinate analysis (PCoA)

analysis showing different beta diversity between groups. (b) The analysis of similarity (Anosim)

was applied to evaluated significant separation of the gut microbiota of the FMT-mice.

**Q6.** pg 8, line 11: Please specify "potentially" pathogenic as pathogenicity was not

determined.

We have revised that content as follows below. (1) At genus level, the

abundances of several potentially facultative pathogens such as *Enterococcus* and

*Escherichia Shigella* were increased, whereas *Akkermansia* and *Bacteroides* were

decreased in FMT-HTGP mice. (Page 8 lines 7 to 9) (2) Recent study revealed that HTGP

patients exhibit lower diversity of gut microbiota and increased abundance of

potentially pathogenic bacteria, such as *Enterobacteriaceae* and *Enterococcus*, which

are associated with disease severity and poor prognosis in HTGP patients. (Page 4 lines

7 to 10)

**Q7.** pg 8, line 19: Pathogen colonization was not tested in this study. Please clarify.

To determine the role of HTGP-altered gut microbiota in pancreatic injury, the
human fecal samples were collected and transplanted into mice. Initially, mice were
reared in SPF condition and treated with broad-spectrum antibiotics to deplete
endogenous gut microbiota. Antibiotic treatment caused a conspicuous enlargement
of ceca and dramatically decreased the α -diversity of gut microbiota (Supplementary
Fig.2a,b, Fig.R4a,b). (Page 6 lines 20 to 22)

Next, the fecal 16S rDNA copy numbers were assessed, suggesting FMT
reconstituted the enteric bacterial abundance before HTGP induction (Supplementary
Fig.2c, Fig. R4a). The human fecal bacteria were labeled by fluorescent dye CM-Dil,
and the red fluorescent signals can be successfully detected. Fluorescent signals were
also observed in feces from recipient mice on 7 days after intragastric administration,
indicating donors' gut bacteria could be transplanted and colonized in recipient mice
(Supplementary Fig.4a, Fig.R4d). In addition, 16S sequencing revealed that more than
50% bacterial species (relative abundance>0.001) identified from healthy volunteers
and HTGP patients, respectively, were colonized in the recipient mice (Supplementary
Fig.4b, Fig.R4e). These results suggest that fecal bacteria from donors were
successfully colonized within the intestinal tract of recipient mice, and validate the
reliability of FMT. (Page 7 lines 14 to 20)

**Fig.R4 FMT reshapes gut microbiota in recipient mice.** (a) Cecum dissected from PBS-treated or
 ABX-treated mice. (b) Alpha diversity of PBS-treated and ABX-treated mice were measured. (c) 16S
 rDNA copies per gram of feces as detected by qRT-PCR. (d) Staining of live bacteria in fecal samples
 from donors and FMT-recipient mice. (e) Species detected in donors and recipient mice by Venn
 diagram.

**Q8.** pg 9, lines 7-10: What data do the authors' have to support these statements?

In current study, potential metabolic pathways modulated by different gut
 microbiota were mapped by PICRUSt analysis (pg 9, lines 7-10). All findings were built
 beyond the altered gut microbiota in FMT mouse models. It's not precise to describe
 that gut microbiota promotes the initiation of immune response and impacts HTGP
 development. Next, we conducted metabolomic and immune experiments to support
 our conclusions. We agreed with your previous suggestion and we have removed the
 PICRUSt results in the revised manuscript.

**Q9.** pg 15, line 1: What is PMA?

The 12 phorbol 13-myristate acetate (PMA) is an artificial stimulus to profoundly
induce neutrophil production of extracellular traps^{6,7}. Previous studies have
demonstrated that both autophagy and superoxide generation are essential for NETs
formation⁸. To determine whether taurine can inhibit NETs, isolated neutrophils were
stimulated by 100 nM 12 phorbol 13-myristate acetate (PMA) and treated by taurine.
We found that PMA-induced NETs were indeed restrained by taurine. We have added
the full name of PMA when it was firstly used. (Page 13 line 14)

**Q10.** pg 22, line 14: What number(s) were samples normalized to?

Data normalization was conducted with a standard of sequence number
corresponding to the sample with the least sequences. For human fecal samples, the
standard number of sequencing tags was 42,572. For recipient mice faeces, the
standard number of sequencing tags was 39,919.

**Q11.** pg 25, line 6: What are the demographics of the donors? Were they matched?

In the fecal microbiota transplantation study, the fecal samples of HTGP patients
were randomly collected, and we then collected six HTG controls and six healthy
volunteers with matched general characteristics. The demographics of donors were
shown in Supplemental Table 2. The baseline of clinical characteristics had no
difference except for BMI and triglycerides levels. The six HTGP patients and six HTG
patients exhibited similar BMI (28.18 ± 1.44 vs 28.82 ± 3.10 , $P=0.6595$) and

1 triglycerides (9.03 ± 2.58 vs 9.59 ± 2.73 , $P=0.7234$) levels. We have added the
 2 demographics of the FMT donors in Supplementary Table 2. (Page 22 line 24)

Supplementary Table 2. Demographic and clinical characteristics of FMT donors.				
Characteristics	HC(n=6)	HTG(n=6)	HTGP(n=6)	P value
Male, n (%) [*]	3(50.00)	3(50.00)	3(50.00)	1.000
Age(years), mean (SD) [#]	41.83(±7.94)	43.83(±7.47)	42.50(±8.46)	0.902
Triglyceride(mmol/L),mean (SD) ^{##}	1.70(±0.81)	9.03(±2.58)	9.59(±2.73)	0.003 [†]
BMI (kg/m ²), mean (SD) ^{##}	24.70(±2.78)	28.18(±1.44)	28.82 (±3.10)	0.046 [†]
Drinking, n (%) [*]	3(50.00)	2(33.33)	3(50.00)	1.000
Smoking, n (%) [*]	2(33.33)	2(33.33)	1(16.67)	1.000
Hypertension, n (%) [*]	1(16.67)	2(33.33)	2(33.33)	1.000
Diabetes, n (%) [*]	4(66.67)	3(50.00)	3(50.00)	1.000

3 ^{*} Fisher's exact test; [#] One-Way ANOVA; ^{##} Nonparametric rank sum test

[†] Indicated HC group was significantly different from the other two groups.

**Q12.** pg 25, line 9: How much material was used for each gavage (vol and number of
 cells)?

Fecal microbiota suspensions were prepared by adding 1 g in 10 ml of sterile
 PBS, shaken for 3 min, and then filtered through a 100 μ m pore mesh. The final
 bacterial suspension was stored at -80 °C until use. After seven days of antibiotic
 treatment, the solution was orally gavaged to the recipient mice (200 μ L each recipient,
 three times a week for three weeks). *Bacteroides uniformis* was cultured using an
 anaerobic chamber, according to the manufacturer's instructions. *Bacteroides*
 *uniformis* (1×10^8 CFU /200 μ L per mouse) was administrated by oral gavage every other
 15 day for 3 weeks. (Page 23 lines 1 to 6, Page 25 lines 16 to 17)

**Q13.** Fig 1, panel D: Please correct axis label to ASV for consistency.

We appreciate your correction and have changed the axis label (Fig.2d, Fig. R5).

**Fig.R5** Alpha diversity, as revealed by shannon index.

**STORMS checklist:**

**Q1.** 1.2: The type of sequencing is not described in the abstract.

We have added "16S rRNA sequencing" in the abstract of STOTMS checklist.

**Q2.** 1.3: Sample types should be described in the abstract.

Microbial profiling by 16S rRNA gene amplicon sequencing was performed on
fecal samples in our study. We have added the description of sample types in the
abstract.

**Q3.** 3.6: Description of the sample size is not justified.

In human cohort, the fecal samples were collected from 25 HTGP patients, 8
serum triglycerides-matched controls and 16 healthy volunteers. The exclusion criteria
included chronic pancreatitis, inflammatory bowel disease, cancer, irritable bowel
syndrome, gastroenteritis, and use of antibiotics, probiotics or laxatives within two
19 months of enrollment. The fecal DNA of 49 stool samples were extracted and
20 sequenced. For the FMT experiment, the fecal samples of mice (FMT-HC group (n=6),

FMT-HTG group (n=8) and FMT-HTGP group (n=8)) who have received fecal microbiota
transplantation from HTGP, HTG and healthy donors were collected, and 16S rRNA
sequencing was performed to validate the fidelity of the microbiota transplantation.
Thus, the sample size of this study was not equal.

**Q4. 4.8:** Negative controls not described for PCR reactions.

We have added the description of PCR protocol for negative controls in 4.8. In
negative control group, the ddH₂O was used for PCR amplification.

**Q5. 4.10:** Mouse housing needs to be described to understand rigor of replication.

In our study, all animals were housed in specific pathogen-free environment with
controlled conditions, including a constant temperature ($22 \pm 2^{\circ}\text{C}$), 12-h light/12-h
dark cycle and $50 \pm 5\%$ humidity. The animals had free access to water and standard
chow (Xie Tong Biotechnology, SFS9112). The drinking water was replaced twice a
13 week. The above animals were acclimated for one week before starting the
14 experiment. For the FMT experiment, fresh fecal samples were collected from six
HTGP patients, six HTG controls and six healthy volunteers. Fecal slurry from each
donor was colonized to three mice, and three recipient mice from each donor were
housed in the same cage to avoid cross contamination.

**Q6. 6.0:** Discussion should acknowledge that only male mice were used.

All the wild type mice used in fecal microbiota transplantation study were from
same gender (male). The gene knockout mice, including GPIHBP1^{-/-} mice and PAD4^{-/-}
mice, were employed by sex to allow an approximately equal number of females and
males. Age- and sex-matched wild-type mice were used as controls. Thus, this finding
applies to both sexes.

**Q7. 7.1:** 16S rRNA amplicon processing should also be noted in checklist

The 16S rRNA amplicon processing has been added in the 7.1.

**Q8.** 8.1: Raw data must be deposited in a public repository and an accession number(s)
provided. <https://www.nature.com/sdata/policies/repositories>

The raw data for generating figures have been uploaded as Source Data file. The
16S rRNA gene sequences were provided and available at National Center for
Biotechnology Information Sequence Read Archive (SRP) database with accession
code PRJNA950569. The metagenomics data, RNA sequencing and single-cell data
have been deposited in the Genome Sequence Archive in National Genomics Data
Center, China National Center for Bioinformation/Beijing Institute of Genomics,
Chinese Academy of Sciences (BioProject: PRJCA016154). Source data are provided
within manuscript.

**Q9.** 8.2; The current statement in 8.1 applies to 8.2

We have revised 8.1 to 8.2 in the revised checklist.

**Q10.** 13.0/1: Limitations and bias need to be addressed.

In our study, the fecal samples were from a single center. Multicenter studies with
a larger sample size are required for validating our findings. We have added the
limitations in the checklist.

**Reference**

1. Pan, Q., et al. Elderly Patients with Mild Cognitive Impairment Exhibit Altered Gut
Microbiota Profiles. *Journal of immunology research*, **22**, 5578958 (2021).

2. Noval Rivas, M., et al. A microbiota signature associated with experimental food
allergy promotes allergic sensitization and anaphylaxis. *The Journal of allergy and*
*clinical immunology*, **131**, 201–212 (2013).

- 3. Kartal, E., et al. A faecal microbiota signature with high specificity for pancreatic
cancer. *Gut*. **71**, 1359–1372 (2022).
- 4. Zou, M., et al. Gut microbiota on admission as predictive biomarker for acute
necrotizing pancreatitis. *Frontiers in immunology*. **13**, 988326 (2022).
- 5. Hu, X. et al. Variations in gut microbiome are associated with prognosis of
hypertriglyceridemia-associated acute pancreatitis. *Biomolecules*. **11**, 695 (2021).
- 6. Remijsen, Q., et al. Neutrophil extracellular trap cell death requires both
autophagy and superoxide generation. *Cell research*, **21**, 290–304 (2011).
- 7. Li, G., et al. Microbiota metabolite butyrate constrains neutrophil functions and
ameliorates mucosal inflammation in inflammatory bowel disease. *Gut microbes*, **13**,
1968257 (2021).
- 8. Yang, L. Y., et al. Increased neutrophil extracellular traps promote metastasis
potential of hepatocellular carcinoma via provoking tumorous inflammatory
response. *Journal of hematology & oncology*, **13**, 3 (2020).

**Reviewer #2**

Gut microbiota aggravates neutrophil extracellular traps-induced pancreatic injury
in hypertriglyceridemic pancreatitis. The authors show on the basis of human
volunteers and patients that gut dysbiosis is present in hypertriglyceridemic
pancreatitis patients (HTGP). Especially *Bacteroides uniformis* is reduced. The authors
also find a correlation between the presence of *Bacteroides uniformis* and the
metabolite taurine and can show in animal studies that taurine is able to suppress net
formation of neutrophils and thus reduce the severity of the disease. Recent studies
clearly show a dysbiosis in patients with acute and chronic pancreatitis, especially

facultative pathogens increase and displace supposedly protective strains such as
*Bacteroides uniformis*. However, why patients with HTGP were used and not acute
pancreatitis patients in general is not clear to me. The mechanism described is
probably not specific for HTGP patients. It is widely known that BMI is associated with
serum triglycerides and changes of the gut microbiome. A more general approach to
the role of the microbiome would be more useful and a general interest. A detailed
breakdown of the mechanism would also significantly strengthen the manuscript.
Besides neutrophils, monocytes play a much more important role in pancreatic
damage.

Thank your overall views and constructive suggestions. Following all comments,
we firstly added a new patient cohort which included other types (mainly alcoholic
and biliary associated) of AP patients. The gut microbiota composition between HTGP
patients and other subtypes of AP patients were compared, and FMT experiments
were then performed. Next, additional works have been done to further illustrate the
proposed mechanism of understanding how *B. uniformis*-produced taurine restrains
NETs formation and alleviates HTGP. Our point-by-point responses are listed below.

**Major points:**

**Q1.** The authors compare healthy volunteers (HC), high triglycerides (HTG) and
hypertriglyceridemic pancreatitis patients (HTGP) with regard to their faecal
microbiome. However, in order to draw conclusions about the effect of the
microbiome on pancreatitis, especially the effect of high triglycerides, the comparison
to the microbiome of patients with acute pancreatitis without hypertriglyceridemia is
missing.

The rapid economic growth and a shift in diet structure cause higher incidence
and increased mortality of hypertriglyceridemic pancreatitis (HTGP) that has
surpassed alcohol as the second cause of AP¹⁻³. Accumulating evidence indicates that
HTGP patients account for higher risks of infected pancreatic necrosis (IPN), organ
failure, longer hospitalization, and higher mortality. The “gut-pancreas axis” is
involved in a variety of pancreatic disorders, including tumors, diabetes and
pancreatitis. Thus, investigating the pathogenetic role of altered gut microbiota in AP
patients may help to find more clues of gut microbiota dysbiosis-induced pancreatic
injury and systemic inflammation.

As suggested, a discovery cohort was added to explore different gut microbiota
composition between HTGP patients and non-hypertriglyceridemia AP patients (HTGP
patients, n=25; non-hypertriglyceridemia AP patients (AP), n=20). Consistent with
previous findings, our findings revealed that HTGP patients exhibited decreased
microbial diversity, and beta-diversity analysis demonstrated a clear distinction
between HTGP patients and other types of AP patients (Supplementary Fig.3a,b and
Fig.R6a,b). In current study, both HTGP patients and AP patients exhibited imbalanced
gut microbiota, including increased abundance of potentially facultative pathogens
such as *Enterococcus* and *Escherichia-shigella*. Notably, higher abundance of
*Escherichia-Shigella* and *Klebsiella*, and lower abundance of *Bacteroides* were found
in HTGP patients compared with non-hypertriglyceridemia AP patients
(Supplementary Fig.3d and Fig.R6c). Our data revealed that the abundance of
*Bacteroides uniformis* was significantly decreased in HTGP patients (Supplementary
Fig.3c and Fig.R6d), suggesting the disordered gut microbiota can be caused by

overrepresented opportunistic pathogenic and decreased beneficial species such as
 *Bacteroides uniformis* in HTGP. (Page 7 lines 5 to 13)

 **Fig.R6 Alterations of gut microbiota in healthy volunteers, HTGP patients and AP patients (non-**
 **HTG AP patients).** (a) Alpha diversity between AP patients and HTGP patients (based on chao1 and
 ACE, AP, acute pancreatitis patients without hypertriglyceridemia, n=20; HTGP,
 hypertriglyceridemic pancreatitis, n=25). (b) Principal coordinate analysis (PCoA) analysis showing
 beta diversity using unweighted unifrac metric distance. (c) The hierarchical clustering heatmap of
 gut bacteria at genus level between HTGP patients, AP patients and healthy volunteers. Top 10
 abundant bacterial genus were compared. (d) The abundance of *Bacteroides uniformis* between
 AP and HTGP patients (AP, n=20; HTGP, n=25).

 **Q2.** Figure 2 shows nicely the animal experiments and the faecal transfer from
 patients. Here I am a bit confused. Authors proclaim 3 mice by each donor (6/group).
 So, 18 animals per group, but the graphs show only 6-8 mice in one group, why were

more than half of the mice excluded? Here also controls missing without pancreatitis
induction. Does the simple Transfer of microbiome affects the organ?

First, we would like to explain why only 6-8 mice were shown in each group. In
our FMT study, fresh fecal samples were collected from six HTGP patients, six HTG
controls and six healthy volunteers, and fecal suspension from each donor was
colonized to three mice. We firstly performed FMT experiments to explore the role of
altered gut microbiota in HTGP, and 6-8 mice per groups were further used for
exploring the changes of gut microbiota and serum metabolites. Only those mice with
matched 16s rRNA sequencing and serum metabolomic data were included in Figure
2 (Fig.2a-c). Here, all the quantification data of 18 mice per FMT-group was
complemented in the Rebuttal letter (Fig.R7a), which exhibited exactly same
phenotype. Compared with FMT-HC and FMT-HTG mice, FMT-HTGP mice displayed
aggravated pancreatic inflammation, including higher serum amylase, higher serum
lipase, higher levels of pro-inflammatory cytokines and increased histologic score
(Fig.R7b-d). No difference was found between FMT-HC and FMT-HTG mice, suggesting
that bacteria-induced deterioration is determined by HTGP-dependent pancreatic
injury rather than hypertriglyceridemia-caused metabolic disorder. (Page 7 lines 2 to
5)

For answering the second question about the missing of controls, we had the
control group when we performed this study (Fig.R7b). To simplify the experiment, the
control group was not included in the figure. As suggested, the extra controls without
pancreatitis were added.

The third question was aimed to explore whether FMT affects the function of
pancreas or whether FMT modulates bacteria translocation to pancreas. The crucial

roles of gut-pancreas axis have been highlighted in AP development. Transplantation
of microbiota from heparanase-exacerbated mice to wild type recipient mice
exacerbates AP, and Hpa-exacerbated AP is alleviated by transfer of the microbiota
from wild type mice⁴. In addition, recolonization of the gut microbiota by FMT
exacerbates AP-induced gut disorders and increases the severity of AP⁵. Our previous
study has revealed that normobiotic FMT alleviates AP-induced gut microbiota
dysbiosis and ameliorates the severity of AP, including mitochondrial dysfunction,
oxidative damage and inflammation⁶. Furthermore, through human-into-mice FMT
experiments from short-term survivors and long-term survivors from pancreatic cancer,
FMT modulates tumor microbiome landscape and affects tumor growth⁷. To further
explore whether FMT can affects bacterial load in the pancreas, RNA fluorescence in
situ hybridization (FISH), with a universal probe against bacterial 16S rRNA, was used
to detect bacteria signals. We found increased bacteria signals in the pancreas of HTGP
mice. However, the quantification data showed no difference between FMT-HC and
FMT-HTGP mice, suggesting HTGP determines the gut microbiota translocation, rather
than specific bacterial species obtained from donors (Fig.R8).

1

2 **Fig.R7 Gut microbiota plays a causal effect on HTGP progression. (a) FMT experimental design,**

C57BL/6 mice were colonized with gut microbiota obtained from HC, HTG and HTGP donors (n=6).
(b) Representative images of H&E staining in control and HTGP groups. (c) Representative images
of pancreas between different FMT groups. (d) Quantification of histology score (Scale bar=100
μ m). (e) Serum amylase, lipase, TNF- α and IL-1 β levels (HTGP, n=6; FMT-HC, n=18; FMT-HTG, n=18;
FMT-HTGP, n=18).

**Fig.R8 Bacteria detection in pancreatic tissue.** The bacterial 16S rRNA sequences in the pancreas
was detected by fluorescence in situ hybridization (FISH) (Scale bar=50 μ m, green). Cell nuclei
stained with 4',6-diamidino-2-phenylindole (DAPI).

**Q3.** Does the FMT affects intestinal barrier and immune response? Th17/Treg axis is
essential for the immunological arm of the intestinal barrier, and it is known that IL17A
is released from Th17 cells and ILC3s in the lamina propria and is able to induce NET
formation (PMID: 26964500).

Gut microbiota disrupts intestinal barrier function and immune system, which are
considered as determinate effects on acute pancreatitis progression. We have
explored the impacts of HTGP on interfering gut barrier function, and the expressions
of Claudin-1, Occludin and ZO-1 were decreased in intestinal epithelium, suggesting
that HTGP impairs gut barrier which may drive bacteria translocation (Supplementary
Fig.6c and Fig.R9a). Interestingly, we found no difference in intestinal barrier injury
between FMT-HC, FMT-HTGP and HTGP mice (Supplementary Fig.6d,e and Fig.R9b),

indicating that HTGP causes increased gut permeability that allows bacteria and
produced taurine to enter circulation and involve in pancreatic damage. (Page 9 lines
22 to 24)

Next, we investigated the effects of FMT from HC donors and HTGP patients on
immune modulation in lamina propria. We observed that FMT-HTGP increased the
proportions of CD4⁺/IL-17A⁺ Th17 cells ($P<0.001$) and improved the Th17/Treg ratio in
lamina propria (Supplementary Fig.6f and Fig.R9c), suggesting HTGP-associated gut
microbiota induced Th17-related pro-inflammatory response. However, the
proportion of ILC3s in lamina propria was found no difference between two groups
(Fig.R9d). Previous studies have described that IL-17A induces pancreatitis through
formation of Padi4-dependent neutrophil aggregates⁸⁻¹¹. In our study, the expressions
of IL-17A in the pancreas were decreased in FMT-HC mice compared with FMT-HTGP
mice, as well as in taurine-treated mice compared to control mice (Fig.3a,
Supplementary Fig.9g and Fig.R10a,b). Our results suggest that colonization with gut
microbiota from HTGP patients causes lower *B.uniformis* abundance and reduces
taurine level, which lead to the imbalance of intestinal Th17/Treg ratio and promote
the intestinal IL-17A release. IL-17 may further travel to pancreas and promote NETs
formation. (Page 10 lines 3 to 6)

**Fig.R9 FMT affects intestinal barrier and immune response.** (a) Western blot analysis showing the

expressions of intestinal epithelial tight junction proteins ZO-1 and Occludin between Control and

HTGP mice. (b) Representative images of immunofluorescence in colon tissues with antibodies

against ZO-1 and Claudin-1. (c) Representative staining and statistical analysis of IL-17A⁺CD4⁺ T cells

and FOXP3⁺ CD4⁺ T cells subsets in colonic lamina propria. (d) Representative staining and statistical

analysis of RORyt⁺ CD3⁻ ILC3 cells in colonic lamina propria.

**Fig.R10 The expression of IL-17A in the pancreas.** (a) The mRNA expressions of IL-17A in the
 pancreas between FMT-HC and FMT-HTGP mice. (b) Quantitative RT-PCR and ELISA tested
 expressions of IL-17A in the pancreas between Tau+HTGP and HTGP mice.

**Q4.** What exact test was used for Figure 2C. The standard deviation is quite high, what
 correction method was used for multiple testing. This point is not very clear in
 Material and Methods.

We have described the statistical methods much details in the revised manuscript.
 For mouse study, One-way ANOVA with the Tukey post hoc test was performed for
 multiple comparisons test. The “statistical analyses” in the “materials and methods”
 section was revised. (Page 30 lines 16 to 21)

**Q5.** Figure 2C shows LPS which is not mentioned in the text or the figure legend. Did
 the authors proclaim that the animals develop an infected necrosis? If yes, then this
 point need to be clarified by additional experiments to show which bacteria infiltrate
 the pancreas.

In our study, LPS is the abbreviation of lipase. Basically, serum lipase was
 measured to determine the role of HTGP-altered gut microbiota in pancreatic injury.
 We have changed all “LPS” to “lipase” in figure and figure legends to avoid any

confusion (Fig.2c, Fig.4g, Fig.6b, Supplementary Fig.8c and Fig.R11a). Since AP model,
rather than SAP model, was used in current study, mice will not get any infected
necrosis in the pancreas (Fig.5g and Fig.R11b). (Page 7 line 4)

**Figure R11.** (a) Serum Lipase levels. (b) Representative photos and images of pancreas by H&E
staining (Scale bar=100 μm).

**Q6.** In general, the FMT experiments with faecal samples from patients with acute
pancreatitis without hypertriglyceridemia are missing and would strongly support the
story.

In clinic, HTGP patients have a higher severity and account for higher risks of SAP
than other types of AP patients. Our data showed that both HTGP and AP patients
exhibited gut microbial dysbiosis. We then conducted FMT experiments to explore the
different pro-inflammatory effects of gut microbiota obtained from HTGP (FMT-HTGP)
and other types of AP patients (FMT-AP). Our results revealed that FMT-HTGP caused
aggravated pancreatic injury, such as higher levels of pancreas histological score,
lipase and trypsin activity, compared with FMT-AP in GPIHBP1^{-/-} mice, suggesting that

HTGP-associated gut microbiota dysbiosis is a crucial driver for accelerating HTGP
 development (Supplementary Fig.3e,f and Fig.R12a,b). (Page 7 lines 8 to 13)

 **Figure R12. FMT experiments using fecal samples from HTGP patients (FMT-HTGP) and other**
 **types of AP patients (FMT-AP).** (a) Representative images of pancreas by H&E staining (Scale
 6 bar=100 μm) and quantification of Ly6g positive cells by IHC (Scale bar=50μm). (b) Serum amylase,
 lipase, IL-1β levels and trypsin activity (HTGP, n=6; FMT-AP, n=6; FMT-HTGP, n=6).

 **Q7.** Authors found a more pronounced neutrophile and monocyte/macrophage
 infiltration in the pancreas of FMT-HTGP mice, both are increasing in the same manner.
 Why authors focus on neutrophils instead of macrophages. It is known than
 macrophages are key cells for inflammatory response in pancreatitis (PMID:
 32739869). On the other hand, it is known that neutrophils can induce intrapancreatic
 protease activation via release of ROS (PMID: 11910350), could authors measure
 increased trypsinogen activation in the pancreas of FMT-HTGP mice?

Neutrophils and macrophages are main immune populations that are involved in
the pathophysiology of acute pancreatitis¹². Our results found FMT-HTGP induced
neutrophils and macrophages infiltration in HTGP mouse models. The RNA seq and
scRNA seq were performed to further dissect the mechanisms of gut microbiota
influencing HTGP development. In RNA seq data, FMT-HTGP associated genes were
enriched in cell movement of neutrophils pathway. Genes increased in FMT-HTGP
mice were involved in inflammatory responses, including leukocyte activation,
neutrophil infiltration, neutrophil degranulation and neutrophil extracellular traps
formation (Fig.3b,d and Fig.R13a,b). The single cell sequencing analysis indicated that
FMT-HTGP mice had higher percentage of neutrophils compared with macrophages
in the pancreas. In FMT-HTGP mice, the proportion of neutrophils was almost 6 times
higher than that in FMT-HC mice (Fig.3f and Fig.R13c). Thus, our study mainly focused
on the roles of gut microbiota on neutrophils modulation. (Page 9 lines 1 to 16)

Next, the trypsinogen activation was measured in different mouse models¹³⁻¹⁵.
Compared with FMT-HC mice, FMT-HTGP mice exhibited increased trypsinogen
activation in the pancreas ($P<0.05$). In addition, *B.uniformis* and taurine
supplementation significantly decreased the trypsin activity in GPIHBP1^{-/-} mice ($P<0.05$)
(Fig.6e, Supplementary Fig.7g, Supplementary Fig.5d and Fig.R14). (Page 29 lines 10 to
12)

**Fig.R13 Gut microbiota dampens HTGP progression by suppressing neutrophil extracellular traps**

**formation.** (a) Different genes enriched in leukocyte activation, neutrophil infiltration and

neutrophil degranulation pathway between FMT-HC and FMT-HTGP mice (FMT-HC, n=3; FMT-HTGP,

n=3). (b) GSEA snapshots of neutrophil extracellular traps formation pathway enrichment analysis

and different expressed genes enriched in NETs formation pathway. (c) The proportion of

different cell types between FMT-HC and FMT-HTGP mice.

**Figure R14. The levels of trypsin activation in the pancreas.**

**Q8.** Figure 5 J-M is missing in the result section. Figure 5K shows flow cytometry of

neutrophils isolated from the pancreas. It seems that Taurin treated mice have a much

higher number of leukocytes which leads to a lower percentage of neutrophils. What

is about macrophages and monocytes? Additional plot with the cell count would verify
the results.

We have revised the results section matched with Figure 5 J to M as following.
We then explored the effects of taurine on alleviating HTGP in GPIHBP1^{-/-} mice,
indicating that taurine supplementation reduced neutrophils infiltration and NETs
formation. These suggest that taurine can limit HTGP exacerbation, regardless of the
mechanisms of causing hypertriglyceridemia. (Page 12 lines 19 to 22, Page 13 lines 1
to 2)

As shown in revised Figure 5k, the flow cytometry was repeated, and taurine
supplementation decreased the proportion of neutrophil in the pancreas (21.36% ±
6.03% vs 7.76% ± 1.94%, respectively, $P < 0.01$) (Fig.5l and Fig.R15). The CD11b⁺Ly6g⁺
cell counts were added in the figure 5k (1148/4831 cells vs 359/4202 cells,
respectively). Our results indicated that the percent of monocytes ($P = 0.0528$) and
macrophages ($P < 0.05$) were both decreased in the pancreas after taurine
administration (Supplementary Fig.10a and Fig.R16a,b). (Page 12 lines 20 to 21)

**Figure R15.** Representative plots of flow cytometry and statistical analysis indicating the
percentage of neutrophils in the pancreas.

**Figure R16.** Representative plots of flow cytometry and statistical analysis indicating the
 percentages of monocytes and macrophages in the pancreas.

**Q9.** Can the authors show that significant amounts of taurine reach the pancreas?

We have compared the concentrations of taurine in the pancreas between
 Tau+HTGP and HTGP groups, as well as FMT-HTGP and FMT-HC groups using metabolic
 analysis and ELSIA respectively. Targeted metabolomic profiling revealed that the
 levels of the taurine in the pancreas were significantly upregulated in Tau+HTGP group
 compared with HTGP group ($1280 \pm 105.50 \mu\text{g/g}$ vs $819.2 \pm 54.96 \mu\text{g/g}$, respectively,
 $P < 0.0001$) (Supplementary Fig.9d and Fig.R17a). In the FMT study, FMT-HC increased
 taurine level in the pancreas compared with FMT-HTGP ($110.50 \pm 24.50 \text{ ng/ml}$ vs 69.1
 $\pm 25.23 \text{ ng/ml}$, respectively, $P < 0.05$) (Fig.R17b). These suggest that both taurine
 supplementation and FMT-HC improve systemic taurine level that can also be
 delivered to the pancreas. (Page 12 line 22)

**Figure R17. Taurine levels in the pancreas.** (a) Targeted metabolite profiling indicating increased
taurine level in the pancreas in Tau+HTGP group compared with HTGP group. (b) ELISA showing
increased taurine level in the pancreas of FMT-HC mice compared with FMT-HTGP mice.

**Q10.** Disease severity in nearly all animal experiments is mainly shown by histological
scoring. Could the authors also provide data such as amylase and lipase activity in
serum, intrapancreatic protease activation as well as lung Mpo as marker for systemic
inflammation. The disease severity is the key point in argumentation and need to be
verified by different methods not only histology score. A clear evaluation of systemic
immune response would help to understand the results better and would give an idea
how NET formation influence disease severity.

We have examined the levels of amylase, lipase activity, intrapancreatic protease
activation, as well as the Mpo levels in the lung, across all the mouse models. FMT-
HTGP caused higher levels of serum amylase, lipase and proinflammatory factors, and
increased intrapancreatic trypsin activation, indicating that disrupted gut microbiota
by HTGP exacerbated systemic inflammation in HTGP (Fig.2c, Supplementary Fig.6a,b
and Fig.R18a-c). In addition, both *B.uniformis* and taurine decreased the levels of
serum amylase, lipase, intrapancreatic trypsin activation, TNF- α , IL-1 β levels, as well
as the Mpo level in the lung (Supplementary Fig.10b-d and Fig.R19a-c). The systemic
inflammatory levels were consistent with the histopathological analysis, suggesting
that HTGP-derived gut microbiota induced pancreatic inflammation, lung injury, and
systemic inflammatory status. (Page 7 lines 2 to 5)

To better understand the roles of gut microbiota on NETs formation and systemic
immune response, the immune profilings in the spleen were explored using flow

cytometry. We observed that taurine pretreatment significantly inhibited the
 proliferation of neutrophils and macrophages (Supplementary Fig.10e and Fig.R19d).
 Taken together, these results suggest that *B.uniformis* and taurine inhibit NETs
 formation that alleviate pancreatic injury and systemic inflammation. (Page 12 lines
 20 to 22)

 **Figure R18. FMT aggravates pancreatic injury and systemic inflammation in HTGP.** (a) The levels
 of amylase and lipase in the serum . (b) The levels of Mpo activity in the lung. (c) Representative
 images of lung by H&E staining (Scale bar=100 μm) and quantification of Mpo positive cells using
 IHC (Scale bar=50μm).

**Figure R19. Taurine alleviates pancreatic injury and systemic immune response in HTGP.** (a) The
 levels of amylase and trypsin activity in the serum. (b) Representative images of lung by H&E
 staining (Scale bar=100 μ m) and quantification of Mpo positive cells by IHC (Scale bar=50 μ m). (c)
 The activity of Mpo in the lung. (d) Representative plots of flow cytometry and statistical analysis
 showing the percentages of neutrophils, monocytes and macrophages in the spleen.

**Q11.** Taurine is mentioned to reduce NET formation. A mechanism how Taurin acts on
 neutrophils should be shown. In Figure 7F the authors proclaim that Taurin from
 serum inhibits PMA induced NET formation, how does this work? A clear mechanism
 needs to be shown.

Emerging evidence has demonstrated that IL-17 and NF- κ B signaling pathways
 regulate NETs formation. Integrating our findings from microbiome, metabolism and
 transcriptomic analysis, our main hypothesis is that the deficiency of *B.uniformis*-
 derived taurine recruits neutrophils infiltration and activates IL-17 and NF- κ B signaling
 pathways in the neutrophils which induces NETs formation and exacerbates
 pancreatic injury in HTGP. To dissect the mechanisms of taurine suppressing NETs
 release, neutrophils (purity > 99%) were firstly isolated from healthy human peripheral
 blood and treated with PMA to induce NETs. The mRNA levels of NF- κ B and IL-17 and
 protein expressions of NF- κ B signaling pathway were tested. Compared with PMA-
 treated group, pretreatment with taurine decreased the NF- κ B protein expression and
 the phosphorylation of NF- κ B ($P<0.05$) (Fig.7g and Fig.R20a). The mRNA levels of NF-
 κ B ($P<0.01$) and IL-17A ($P<0.05$) were also decreased by taurine treatment (Fig.7h,i
 and Fig.R20b). These suggest that taurine suppresses the activations of NF- κ B and IL-
 17 signaling pathways that can be considered as major mechanisms of *B.uniformis*-
 derived taurine repressing NETs and pancreatic injury in HTGP. (Page 14 lines 16 to 21,
 Page 15 lines 1 to 4)

 **Figure R20. Inhibitory effects of taurine on IL-17 and NF- κ B signaling pathways that suppress**
 **NETs release.** (a) Representative western blot images and quantitative comparison of total and
 phosphorylation levels of NF- κ B protein in neutrophils between PMA group and PMA+Tau group.

(b) The relative mRNA expressions of NF- κ B and IL-17A in neutrophils between PMA group and
PMA+Tau group.

**Q12.** Figure 7G: The results do not support the figure presented here. It is conveyed
that HTGP is induced by neutrophil activation and NET formation, this is simply false.
Neutrophils influence the severity, not the onset of disease. It has been shown that
the IL17-NET axis can lead to ductal obstruction, but for this to happen the cells must
first migrate into the organ and be activated, i.e. the pancreatitis must have already
begun. This figure need to be modified.

We really appreciate your points of NETs-dependent pro-inflammatory roles in
HTGP, rather than the causal effects of HTGP initiation. We have revised the graphic
abstract to give a clear and precise view of our study (Fig.7j and Fig.R21). Basically,
HTGP induces lower abundance of *Bacteroides uniformis* species in the gut microbiota.
Lower abundance of *Bacteroides uniformis* causes the deficiency of taurine production
and accelerates NETs release. On one hand, decreased taurine level accelerates
neutrophil recruitment and NETs release by upregulating NF- κ B and IL-17 signaling
pathways. On the other hand, the impairment of *B.uniformis*-derived taurine
production causes the imbalance of intestinal Th17/Treg cells that enhances intestinal
IL-17A release and promotes NETs formation. As a result, pancreatic injury and
systemic inflammation are aggravated. (Page 18 lines 5 to 13)

 **Figure R21.** Gut microbiota aggravates neutrophil extracellular traps-induced pancreatic injury in
 hypertriglyceridemic pancreatitis

 **Minor points:**

**Q1.** The first sentence of the introduction: and the mortality rate ranges from 20% to
 40%". Such a dramatic mortality rate is only present in severe form of pancreatitis not
 in general.

I really appreciate your comments, and this part has been revised as following,
 "Acute pancreatitis (AP) results from pancreatic enzyme activation and pancreatic
 "self-digestion". The incidence is more than 34 cases per 100,000 general population
 12 per year^{1,2}. As a self-limiting mild disease, approximately 20-30% of AP patients
 develop a severe form, which can cause critical illness and higher mortality rate³. (Page
 3 lines 9 to 12)

**Q2.** Figure 1C. The authors should show the percentage of difference in the NMDS
plot.

As a ranking method to overcome the shortcomings of linear models (including
PCA and PCoA), Non-Metric multidimensional Scaling (NMDS) can better reflect the
nonlinear structure of ecological data^{16,17}. NMDS was used to visualize the differences
of beta diversity between groups. The stress of NMDS less than 0.2 was considered as
a good representation in reduced dimensions. In our study, NMDS analysis revealed
clear separation between three groups (stress=0.139) (Fig.1c and Fig.R22a,b). The
analysis of similarity (Anosim) was used to evaluate the separation of gut microbiota
between HTGP patients and healthy volunteers (R=0.4928, P=0.001). The separation
between HTGP patients and HTG controls was evaluated by Anosim (R=0.17, P=0.044),
suggesting a less heterogeneous community structure between HTGP patients and
controls. (Page 5 lines 9 to 14)

**Fig.R22 The separation of gut microbiota composition.** (a) Nonmetric multidimensional scaling
(NMDS) plot with cluster indicating differential microbial distributions between groups. (b) The

analysis of similarity (Anosim) was applied to evaluated significant separation of gut microbiota in
HC, HTG and HTGP patients.

**Q3.** Can Authors show that the depletion of Gut microbiota in mice worked?

The antibiotics cocktails have been shown to deplete gut microbiota in previous
study⁷ (*Cell*. **178**,795-806, 2019). The protocol used in current study shortened the
exposure time of antibiotic treatment from two weeks to a week. We have validated
the effects of gut microbiota depletion using 16s rRNA sequencing and PCR. We
observed that antibiotics treatment caused a conspicuous enlargement of cecum
(Supplementary Fig.2a and Fig.R23a). The diversity of gut microbiota between pre- and
post-ABX treatment was compared, antibiotic treatment significantly decreased the α -
diversity of gut microbiota and reduced fecal 16S rDNA copy numbers (Supplementary
Fig.2b,c and Fig.R23b), suggesting that antibiotic treatment can clear major gut
resident microbiota prior to FMT. (Page 6 line 20 to 22, Page 7 lines 1 to 2)

**Figure R23. Validation of gut microbiota depletion in mice.** (a) Cecum dissected from PBS-treated
or ABX-treated mice. (b) Alpha diversity of PBS-treated and ABX-treated mice was measured. (c)
16S rDNA copies per gram feces was detected by qRT-PCR.

**Q4.** Co-housing and horizontal microbiome transfer plays an important role in mice
experiments, could the authors explain their strategy.

In our study, mice had free access to water and standard chow diet. The drinking
water bottle was replaced twice a week. Depletion of gut microbiota before fecal
transplantation was achieved by a week of antibiotic treatment (5 mg/ml
streptomycin and 0.1 mg/ml clindamycin) administration through drinking water.
(Page 22 lines 16 to 19)

For the FMT experiment, mice that received FMT from the same donor were
housed in the same cages. Then, mice were randomized divided into control and HTGP
groups. For other experiments, mice in each group were housed at the same cages to
avoid a possible horizontal microbiome transfer. Co-housing of different groups and
experiments in the same cages was forbidden. (Page 22 lines 20 to 24, Page 23 lines 1
to 8)

**Q5.** In Material and Methods, caerulein was administered hourly for 10 h and the
animals were sacrificed 11 h after the first application. In the overview chart (Figure
2A, 4B), however, 1 day after caerulein treatment. This need to be clarified, 12h or
24h after onset of disease makes a difference.

We have revised the overview chart in the manuscript (Figure 1a, Figure 4b,
Figure 5e and Figure 6a). AP model was performed by intraperitoneal injections of
caerulein hourly for 10 h. Mice were euthanized 1h after the ultimate injection (11h
from the initial injection).

**Q6.** Figure 4G, the axis indicate a greater difference than is realistic, the y-axis should
start at 0.

We have adjusted the y-axis from 0 to 200, and added a segment at 120 (Fig.4g
and Fig.R24).

**Figure R24.** Serum lipase levels between HTGP group and B.U+HTGP group.

**Q7.** A detailed explanation of the statistical tests in the material and methods section
is necessary.

Thank you for your valuable suggestion. The statistical analyses in the last
paragraph of the materials and methods section were revised. We hope the revision
could illustrate the methods clearly, which is much easier to be understood.

The error bar represented the standard error of the mean (SEM). Statistical
significance was assigned at $P < 0.05$ and detected by GraphPad Prism 9.0 (GraphPad
Software, San Diego, USA). The Shapiro-Wilk test was used to determine the sample
distribution type. For the comparison of two groups, F test was used to identify the
homoscedasticity, and significance was calculated by the two-tailed Mann-Whitney
test or unpaired, two-sided t -test depending on the sample distribution type. For

comparisons more than two groups, the significance was calculated by the Kruskal-
Wallis test or ordinary one-way ANOVA regarding to the distribution of samples, a post
hoc Tukey test was used to conduct multiple comparisons. Weighted_unifrac and Bray-
Curtis dissimilarity matrices were calculated to produce ordination using PCoA. These
matrices were also used for assessing the differences of β -diversity using Anosim.
PCoA was carried out for all dimension reduction analyses using the vegan package in
R software. Correlations between enriched species and circulating metabolites were
tested with Spearman's correlation. The parametric test (One-Way ANOVA) was used
to compare the difference in clinical data between groups. Categorized variables were
compared using chi-square test or the Fisher exact test, as appropriate. Statistical
analyses were conducted using SPSS v.26.0 (IBM Corporation). All tests were two-
tailed and significance was set at $P < 0.05$. (Page 30 lines 13 to 24, Page 31 lines 1 to 8)

**Reference**

- 1. Boxhoorn, L. et al. Acute pancreatitis. *Lancet*. **396**, 726-734 (2020).
- 2. Schepers, N. J. et al. Impact of characteristics of organ failure and infected necrosis
on mortality in necrotising pancreatitis. *Gut*. **68**, 1044-1051 (2019).
- 3. Simha, V. Management of hypertriglyceridemia. *BMJ*. **371**, m3109 (2020).
- 4. Lei, Y. et al. Parabacteroides produces acetate to alleviate heparanase-exacerbated
acute pancreatitis through reducing neutrophil infiltration. *Microbiome*. **9**, 115
(2021).
- 5. Li, X. et al. The interplay between the gut microbiota and NLRP3 activation affects
the severity of acute pancreatitis in mice. *Gut Microbes*. **11**, 1774-1789 (2020).
- 6. Liu, L.W. et al. Gut microbiota-derived nicotinamide mononucleotide alleviates

acute pancreatitis by activating pancreatic SIRT3 signalling. *Br J Pharmacol.* **180**, 647–
666 (2022).

7. Riquelme, E., et al. Tumor Microbiome Diversity and Composition Influence
Pancreatic Cancer Outcomes. *Cell.* **178**,795-806 (2019).

8. Leppkes, M., et al. Externalized decondensed neutrophil chromatin occludes
pancreatic ducts and drives pancreatitis. *Nature communications.* **7**, 10973 (2016).

9. Zhang, Y. et al. Interleukin-17-induced neutrophil extracellular traps mediate
resistance to checkpoint blockade in pancreatic cancer. *J Exp Med.* **217**, e20190354
(2020).

10. Hagner, M. et al. IL-17A from innate and adaptive lymphocytes contributes to
inflammation and damage in cystic fibrosis lung disease. *The European respiratory*
*journal.* **57**, 1900716 (2021).

11. Li, G. et al. Role of Interleukin-17 in Acute Pancreatitis. *Frontiers in immunology.* **12**,
674803 (2021).

12. Wu, J., et al. Macrophage phenotypic switch orchestrates the inflammation and
repair/regeneration following acute pancreatitis injury. *EBioMedicine.* **58**, 102920
(2020).

13. Merza, M., et al. Neutrophil Extracellular Traps induce trypsin activation,
inflammation, and tissue damage in mice with severe acute
pancreatitis. *Gastroenterology.* **149**, 1920-1931 (2015).

14. Gukovskaya, A. S., et al. Neutrophils and NADPH oxidase mediate intrapancreatic
trypsin activation in murine experimental acute pancreatitis. *Gastroenterology.* **122**,
974–984 (2002).

15. Jancsó, Z., & Sahin-Tóth, M. Mutation that promotes activation of trypsinogen
increases severity of secretagogue-induced pancreatitis in
mice. *Gastroenterology*. **158**, 1083–1094 (2020).

16. Pan, Q., et al. Elderly Patients with Mild Cognitive Impairment Exhibit Altered Gut
Microbiota Profiles. *Journal of immunology research*, **22**, 5578958 (2021).

17. Noval Rivas, M., et al. A microbiota signature associated with experimental food
allergy promotes allergic sensitization and anaphylaxis. *The Journal of allergy and*
*clinical immunology*, **131**, 201–212 (2013).

Thank you for your contributions for reviewing our manuscript. These comments
are all valuable and helpful for improving our manuscript, as well as the important
guiding significance to our further studies. We have tried our best to revise our
manuscript, these changes will not influence the content and framework of the main
study. We appreciate all editors and referees' warm work earnestly, and hope that our
revision will meet with approval for publication.

REVIEWER COMMENTS

Reviewer #1 (Remarks to the Author):

The authors have successfully addressed my comments and the revised manuscript is much improved. One remaining comment on the revised manuscript: pg 5, line 13; pg 8, line 6. Please include the R value for the ANOSIM comparisons.

Reviewer #2 (Remarks to the Author):

The authors did a really good job. The manuscript has been significantly improved. I have no further questions or concerns. Congratulations.

Reviewer #3 (Remarks to the Author):

In this study, Li et al. investigated the role of gut microbiota in HTGP, specifically focusing on its impact on NETs formation and pancreatic injury. The authors found that HTGP patients had reduced gut microbiota diversity and lacked beneficial bacteria. Through fecal microbiota transplantation in mice, they successfully replicated the increased NETs formation observed in HTGP patients, leading to worsened pancreatic injury and systemic inflammation.

The study also delved into the underlying mechanisms, highlighting the significance of *Bacteroides uniformis* in maintaining taurine production and suppressing IL-17 release. Subsequent administration of taurine proved effective in mitigating NF- κ B and IL-17 signaling pathways in neutrophils, resulting in reduced NETs formation and improved pancreatic injury.

Overall, the experiments conducted in this study were comprehensive and logically structured. In addition to two completed initial reviews, I was specifically asked to focus on evaluating the scRNA-

seq part and its implications in immunology. As such, I hope the authors can address the following issues related to this aspect to further strengthen the study's findings.

1. The manuscript lacks a detailed description and plot visualization of quality control for both scRNA-seq and bulk RNA-seq experiments. The number of samples used in each experiment, as well as the transcripts and genes detected in each sample or cell, should be provided in supplemental tables. The sequencing data should be made publicly available.
2. The methods for scRNA-seq need to be heavily elaborated. Specific methods such as functions and parameters used in clustering or module score calculation should be added.
3. The methods for bulk RNA-seq are missing.
4. Genes in figure 3b ,3d, 3h, and 3g should be reported as supplemental tables.
5. Genes in supplemental figure 5b, 5e, and 5f should also be reported in supplemental tables.
6. The authors claimed that neutrophil composition was significantly higher in FMT-HTGP group compared to FMT-HC group (figure 3f). It is important to clarify how the statistical comparison was performed and provide the corresponding p values.
7. In supplemental figure 5b, the method used for ranking pathways should be explained. The authors stated that the plot shows the top 10 pathways, but I see 30 in the plot?
8. The enrichment analysis in supplemental figure 5f appears to have an issue with gene count numbers being extremely low for some pathways. Most enriched pathways had 4 genes covered, and five pathways had only 2 genes covered. I am not sure how could a pathway be enriched if such few genes were encountered in the input list. The same issue applies to supplemental figure 5b. The adjusted p values in supplemental figure 5f seemed to be high for several pathways, how are they included?
9. Despite the issues with scRNA-seq analysis, the data appear interesting. The authors should consider using UMAP if they choose to revise. Additionally, it would be beneficial to show two more tSNE/UMAP plots, with cells colored by conditions and samples, to reveal the variations introduced by different conditions or samples.
10. Based on the tSNE plot, strong heterogeneity exists for many cell types. Sub-clustering of macrophages, neutrophils, B and plasma cells, and T cells could uncover subtype enrichment in different conditions. Especially for the T cells, the authors should try to find out whether there are Th17 sub-clusters, or at least plot the Th17 cytokines, such as IL17A, IL17F, and IL26, to support their Th17 activation hypothesis.
11. It will also be interesting to run cell-cell interaction analysis, especially between the T cells and neutrophils, since they are the major players in the mechanism revealed in the study.

Minor:

1. Figure captions should be elaborated.
2. The authors need to provide clearer information regarding the statistical tests used, and they should report the actual p-values or adjusted p-values.
3. Certain sentences in the manuscript would benefit from rephrasing to improve the clarity and convey information more effectively.

Response to Reviewers

Dear Referees:

Thank you for all referees' comments concerning our manuscript entitled "Gut microbiota aggravates neutrophil extracellular traps-induced pancreatic injury in hypertriglyceridemic pancreatitis (NCOMMS-23-01234A)". We have carefully studied all comments and made revisions in accordance with the comments. All changes were highlighted with blue color in the revised manuscript. We hope that the revision will be sufficient to make our manuscript suitable for acceptance. The following pages are point-by-point response to each comment.

Reviewer #1 (Remarks to the Author):

Q1.The authors have successfully addressed my comments and the revised manuscript is much improved. One remaining comment on the revised manuscript: pg 5, line 13; pg 8, line 6. Please include the R value for the ANOSIM comparisons.

We appreciate all your helpful suggestion and the R value for the comparisons of ANOSIM have been added in the revised manuscript. (Page 5, line 13; Page 8, line 6.)

Reviewer #2 (Remarks to the Author):

Q1.The authors did a really good job. The manuscript has been significantly improved. I have no further questions or concerns. Congratulations.

We really thank you for all constructive comments and kind evaluation.

Reviewer #3 (Remarks to the Author):

In this study, Li et al. investigated the role of gut microbiota in HTGP, specifically

focusing on its impact on NETs formation and pancreatic injury. The authors found
that HTGP patients had reduced gut microbiota diversity and lacked beneficial bacteria.
Through fecal microbiota transplantation in mice, they successfully replicated the
increased NETs formation observed in HTGP patients, leading to worsened pancreatic
injury and systemic inflammation.

The study also delved into the underlying mechanisms, highlighting the
significance of *Bacteroides uniformis* in maintaining taurine production and
suppressing IL-17 release. Subsequent administration of taurine proved effective in
mitigating NF- κ B and IL-17 signaling pathways in neutrophils, resulting in reduced
NETs formation and improved pancreatic injury.

Overall, the experiments conducted in this study were comprehensive and
logically structured. In addition to two completed initial reviews, I was specifically
asked to focus on evaluating the scRNA-seq part and its implications in immunology.
As such, I hope the authors can address the following issues related to this aspect to
further strengthen the study's findings.

We really appreciate your review and have tried our best to address all your
comments and improve our manuscript. We hope the revision will clarify all your
concerns.

**Q1.** The manuscript lacks a detailed description and plot visualization of quality
control for both scRNA-seq and bulk RNA-seq experiments. The number of samples
used in each experiment, as well as the transcripts and genes detected in each sample
or cell, should be provided in supplemental tables. The sequencing data should be
made publicly available.

Thanks for your kind reminder. In our study, pancreas samples from FMT-HC and
FMT-HTGP mice were harvested and three samples for each group have been
sequenced. As suggested, the detailed description and plot visualization of RNA-seq
quality control were uploaded to the Supplementary table 3 (bulk RNA-seq). For the
analysis of scRNA-seq data, fastQC and fastp were adopted to remove the low-quality
raw reads and adaptor sequences. Next, cells were filtered by UMI counts, gene
counts, and median genes per cell, which were all shown in Supplementary table 4
(scRNA-seq). The sequencing data have been deposited in the Genome Sequence
Archive in National Genomics Data Center, China National Center for
Bioinformatics/Beijing Institute of Genomics, Chinese Academy of Sciences
(BioProject: PRJCA016154). Source data are provided within manuscript.

**Q2.** The methods for scRNA-seq need to be heavily elaborated. Specific methods such
as functions and parameters used in clustering or module score calculation should be
added.

Thank you for your valuable suggestion. The detailed methods have been
described below and added in Supplementary materials.

**Primary analysis of raw read data**

[revised manuscript text omitted]

**Cell-cell communication analysis with CellPhone DB**

Cell-cell interaction (CCI) between T cells and neutrophils were predicted based
on known ligand-receptor pairs by Cellphone DB version. The permutation number to
compute the null distribution of average ligand-receptor pair expression in
randomized cell identities was set to 1,000. Based on the average log gene expression
distribution for all genes across each cell type, a cutoff was defined to establish the

threshold for individual ligand or receptor expression. Predicted interaction pairs with
$p < 0.05$ and of average log expression > 0.1 were considered as significant and
visualized by circlize (0.4.10) R package.

**Q3.** The methods for bulk RNA-seq are missing.

In our study, pancreas tissues from FMT-HC (n = 3) and FMT-HTGP (n = 3) mice
were harvested after developing HTGP model. Total RNAs from three biological
replicates of each group were isolated using RNeasy kit (Qiagen, Shanghai, China), and
total amounts and integrity of RNA were assessed using the RNA Nano 6000 Assay Kit
of the Bioanalyzer 2100 system (Agilent Technologies, CA, USA). The gene expression
profiles of FMT-HC and FMT-HTGP groups were investigated using the Illumina
NovaSeq 6000 according to the manufacturer's guide (Illumina, San Diego, CA, USA).
We have added bulk RNA-seq as part of methods in revised manuscript. (Page 26 line
20 to Page 27 line 3)

**Q4.** Genes in figure 3b ,3d, 3h, and 3g should be reported as supplemental tables.

The raw data for generating figures have been uploaded as Source Data file.
Source data are provided and available as supplemental tables (figure 3b→source data
figure 3b; figure 3d→source data figure 3d; figure 3h→source data figure 3h and figure
3g→source data figure 3g).

**Q5.** Genes in supplemental figure 5b, 5e, and 5f should also be reported in
supplemental tables.

We have uploaded those gene lists as source data which is also available as
supplemental tables (supplemental figure 5b→source data supplementary figure 5b;
source data supplemental figure 5e→source data supplementary figure 5e;
supplemental figure 5f→source data supplementary figure 5f).

**Q6.** The authors claimed that neutrophil composition was significantly higher in FMT-
HTGP group compared to FMT-HC group (figure 3f). It is important to clarify how the
statistical comparison was performed and provide the corresponding p values.

Thanks for your valuable suggestion. Here, we observed increased proportion of
neutrophils in FMT-HTGP pancreatic tissues using flow cytometry ($P<0.05$, Fig.R1a) and
IHC staining ($P<0.001$, Fig.R1b). The single cell sequencing analysis indicated that FMT-
HTGP mice had higher percentage of neutrophils in the pancreas, but no statistical
significance was observed between groups ($P=0.41$) due to the relatively small sample
size. Thus, we have removed the word "significantly" in the revised manuscript. (Page
9 line 14)

**Fig.R1 Gut microbiota dampens HTGP progression by the reduction of neutrophils.** a. Flow
cytometric quantification and statistical analysis of neutrophils infiltration in the pancreas between
FMT-HC and FMT-HTGP groups (n=5). b. Representative images and quantification of Ly6g positive
cells by IHC (Scale bar=50μm; n=6).

**Q7.** In supplemental figure 5b, the method used for ranking pathways should be
explained. The authors stated that the plot shows the top 10 pathways, but I see 30
the plot.

In figure 5b, the ranking pathways were selected based on the order of adjusted
*P* value. Gene ontology (GO) analysis revealed a total of 30 enrichment functions,
including three subsets, biological process, molecular function and cellular component.
Among them, we only focused on the top 10 biological processes enriched by
significantly upregulated genes. Thus, only top 10 biological processes were plotted in
revised figure 5b, and enriched molecular functions and cellular components were all
removed.

**Fig.R2 FMT reduces immune activation in HTGP.** Gene ontology (GO) analysis showing top 10
pathways (biological functions) enriched by up-regulated genes between FMT-HTGP and FMT-HC
groups (FMT-HC, n=3; FMT-HTGP, n=3).

**Q8.** The enrichment analysis in supplemental figure 5f appears to have an issue with
gene count numbers being extremely low for some pathways. Most enriched
pathways had 4 genes covered, and five pathways had only 2 genes covered. I am not

sure how could a pathway be enriched if such few genes were encountered in the
input list. The same issue applies to supplemental figure 5b. The adjusted p values in
supplemental figure 5f seemed to be high for several pathways, how are they included?

We really appreciate your comments. We have reset the threshold for KEGG
analysis (covered gene number ≥ 3 , adjusted p values ≤ 0.05) [5] and found that IL-17
signaling pathway and NF- κ B signaling pathways which can drive NETs formation were
activated in FMT-HTGP group compared to FMT-HC. We have revised Supplementary
figure 5f.

**Fig.R3 Differentially expressed genes in neutrophils between FMT-HTGP and FMT-HC mice.** KEGG
analysis visualizing the upregulated pathways of neutrophils in FMT-HTGP group (covered gene
number ≥ 3).

**Q9.** Despite the issues with scRNA-seq analysis, the data appear interesting. The
authors should consider using UMAP if they choose to revise. Additionally, it would be
beneficial to show two more tSNE/UMAP plots, with cells colored by conditions and
samples, to reveal the variations introduced by different conditions or samples.

As suggested, we have changed “tSNE” plot to “UMAP” plot in figure 3e. To
characterize the effects of gut microbiota on HTGP development, annotated cell types

were visualized with different colours using both UMAP and tSNE algorithms for
 clustering (Fig.R4 and Fig.R5).

 **Fig.R4 Overview of the single-cell transcriptional profiling of cells isolated from pancreas using**
 **UMAP.** a. UMAP visualizing integrated cell types in all samples. b. UMAP visualizing annotated all
 cell types between FMT-HTGP and FMT-HC groups. c. UMAP visualizing individual cell type between
 FMT-HTGP and FMT-HC groups.

 **Fig.R5 Overview of the single-cell transcriptional profiling of pancreatic cells using tSNE.** a. A t-
 SNE plot displaying distinct clusters of 29,082 cells from all samples. b. t-SNE plot displaying
 annotated all cell types between FMT-HTGP and FMT-HC groups. c. t-SNE plot displaying each cell
 type between FMT-HTGP and FMT-HC groups.

 **Q10.** Based on the tSNE plot, strong heterogeneity exists for many cell types. Sub-
 clustering of macrophages, neutrophils, B and plasma cells, and T cells could uncover
 subtype enrichment in different conditions. Especially for the T cells, the authors
 should try to find out whether there are Th17 sub-clusters, or at least plot the Th17
 cytokines, such as IL17A, IL17F, and IL26, to support their Th17 activation hypothesis.

Following your comment, we attempted to extract the T cells population for sub-
 cluster analysis. Since T cells account for a very small population in AP tissue, we failed
 to identify Th17 sub-clusters within T cells (Fig.R6a,b). Regarding to markers and
 activation-related genes of Th17, the expression of IL-17A ($P=0.44$, Fig.R6c) and Rorc
 ($P=0.45$, Fig.R6d) had no difference between FMT-HC and FMT-HTGP mice, and the
 levels of IL-17F and IL-26 were not detected in T cells. Our findings suggest that lacking
 microbiota-produced taurine disrupts the balance of Th17/Treg cells in the gut that
 causes intestinal IL-17A release. Our results emphasize the importance of gut
 microbiota homeostasis on maintaining intestinal immune system which play crucial
 roles in restraining HTGP progression.

 **Fig.R6 Single-cell transcriptional analysis reveals biological characterization of T cells. a.**
 Visualization of T cells using UMAP. b. UMAP visualizing annotated T cell types between FMT-HTGP

1 and FMT-HC groups. c. Visualization of IL-17A level in T cells between FMT-HC and FMT-HTGP mice.

2 d. The comparison of Rorc level in T cells between FMT-HC and FMT-HTGP mice.

**Q11.** It will also be interesting to run cell-cell interaction analysis, especially between
the T cells and neutrophils, since they are the major players in the mechanism
revealed in the study.

Thanks for your valuable suggestion. To explore the crosstalk between T cells and
neutrophils in pancreas, we applied CellPhoneDB, an interactive web application that
infers cellular interactions according to ligand-receptor signaling database^[6]. We found
many significant ligand–receptor pairs between T cells and neutrophils, and T cells
recruited neutrophils through CCL2-CCR2, CXCL10-CXCR3, CCL20-CCR6 and CXCL12-
CXCR4 axes (Fig.R7a-d). However, both FMT-HTGP and FMT-HC mice exhibited similar
types of interaction between T cells and neutrophils and the number and interaction
weight/strength were almost same between two groups. Thus, we speculate that the
gut microbiota-associated crosstalks between T cells and neutrophils are not
determinate factor for exacerbating pancreatic injury.

**Fig.R7 Single-cell transcriptional analysis reveals the cell-cell crosstalks.** a,b. Analysis of the
number and interaction strength between T cells and neutrophils in the FMT-HC and FMT-HTGP
mice. c,d. Identification of signaling through comparing the communication probabilities
modulated by ligand–receptor pairs.

**Minor:**

**Q1.** Figure captions should be elaborated.

Thank you for your suggestion. we have revised figure captions part thoroughly.

**Q2.** The authors need to provide clearer information regarding the statistical tests
used, and they should report the actual p-values or adjusted p-values.

In our study, the *P* values were adjusted using the Benjamini & Hochberg method
and $p_{adj} \leq 0.05$ were set as the threshold for significantly differential expression. The
statistical analyses of scRNA-seq and bulk RNA-seq were added in Materials and
methods section.

**Q3.** Certain sentences in the manuscript would benefit from rephrasing to improve
the clarity and convey information more effectively.

We have finished all the revision, and the manuscript was revised by a native
speaker which we hope to make it much readable and understandable.

Thank you for your contributions for reviewing our manuscript. These comments
are all valuable and helpful for improving our manuscript. We have tried our best to
revise our manuscript, these changes will not influence the content and framework of
the main study. We appreciate all editors and referees' warm work earnestly, and hope
that our revision will meet with approval for publication.

**Reference**

1. Yu, G., Wang, L. G., Han, Y., & He, Q. Y. clusterProfiler: an R package for comparing
biological themes among gene clusters. *OMICS*, **16**, 284–287 (2012).

2. Subramanian, A., et al. Gene set enrichment analysis: a knowledge-based approach
for interpreting genome-wide expression profiles. *Proc Natl Acad Sci U S A*. **102**,
15545–15550 (2005).

3. Cortal, A., Martignetti, L., Six, E., & Rausell, A. Gene signature extraction and cell
identity recognition at the single-cell level with Cell-ID. *Nat Biotechnol*. **39**, 1095–1102
(2021).

4. Andreatta, M., & Carmona, S. J. UCell: Robust and scalable single-cell gene signature
scoring. *Comput Struct Biotechnol J*. **19**, 3796–3798 (2021).

- 5. Zheng, P., Zhang, N., Ren, D., Yu, C., Zhao, B., & Zhang, Y. Integrated spatial
transcriptome and metabolism study reveals metabolic heterogeneity in human
injured brain. *Cell Rep Med.* **4**, 101057 (2023).
- 6. Efremova, M., Vento-Tormo, M., Teichmann, S. A., & Vento-Tormo, R. CellPhoneDB:
inferring cell-cell communication from combined expression of multi-subunit ligand-
receptor complexes. *Nat Protoc.* **15**, 1484–1506 (2020).

REVIEWERS' COMMENTS

Reviewer #3 (Remarks to the Author):

After revision, the results exhibit increased robustness. However, there are still some issues pending further revision:

1. The number of replicates utilized for each condition in the scRNA-seq dataset remains unclear.
2. The description of mapping reads to the GRCh38 (hg38) {GRCm38 (mm10)} transcriptome requires clarification. Were both the human and mouse genomes employed?
3. In the supplementary methods pertaining to DEG analysis, it is advisable for the authors to confirm the implementation of the Bonferroni Correction to control the false discovery rate.
4. The quality of Figure R4 and R5 in the response letter is compromised, with visual blurring. Potential doublets within the scRNA-seq datasets are discernible from the UMAP plot. The authors should consider the removal of these doublets and inserting these plots to supplementary figures.

Response to Reviewer

Dear Reviewer,

Thank you for all referees' comments concerning our manuscript entitled "Gut microbiota aggravates neutrophil extracellular traps-induced pancreatic injury in hypertriglyceridemic pancreatitis (NCOMMS-23-01234B)". We have carefully studied all comments and made revisions in accordance with the comments. We hope that the revision will be sufficient to make our manuscript suitable for acceptance. The following pages are point-by-point response to each comment.

Reviewer #1 (Remarks to the Author):

Q1. The number of replicates utilized for each condition in the scRNA-seq dataset remains unclear.

Thanks for your kind reminder. In our study, three biologically independent animals per group were selected for scRNA sequencing. The description has been added in the figure legend.

Q2. The description of mapping reads to the GRCh38 (hg38) {GRCm38 (mm10)} transcriptome requires clarification. Were both the human and mouse genomes employed?

We apologize for the unclear description. In our study, only the mouse genomes were employed and reads were mapped to the reference genome GRCm38 (mm10) with STAR. We have removed "GRCh38 (hg38)" in the revised manuscript.

Q3. In the supplementary methods pertaining to DEG analysis, it is advisable for the

1 authors to confirm the implementation of the Bonferroni Correction to control the
2 false discovery rate.

To identify DEGs, genes which were expressed in over 10% of total cells from both
groups were selected, and differentially expressed genes with an average log (Fold
Change) value greater than 0.25 were considered as DEGs. In the meantime, adjusted
*P* value was calculated by Bonferroni Correction and the value 0.05 was used as the
criterion to evaluate the statistical significance.

**Q4.** The quality of Figure R4 and R5 in the response letter is compromised, with visual
blurring. Potential doublets within the scRNA-seq datasets are discernible from the
UMAP plot. The authors should consider the removal of these doublets and inserting
these plots to supplementary figures.

Thanks for your suggestion. We have improved the image quality in the
Supplementary figure 5e,f. The potential doublets within the scRNA-seq analysis were
removed, and the further analysis would not be affected. The UMAP and t-SNE
visualization were added into Supplementary figure 5e,f.

Thank you for your contributions for reviewing our manuscript. We have tried our
best to revise our manuscript and we appreciate all editors and referees' warm work
earnestly, and hope that our revision will meet with approval for publication.